# Deciphering an AgRP-serotoninergic neural circuit in distinct control of energy metabolism from feeding

Yong Han[1,2,8], Guobin Xia[1,2,8], Dollada Srisai[3,8], Fantao Meng[1,2,4], Yanlin He [1,2,5], Yali Ran[1,2], Yang He[1,2], Monica Farias[1,2], Giang Hoang[1,2], István Tóth[6,7], Marcelo O. Dietrich [7], Miao-Hsueh Chen[1,2], Yong Xu [1,2] & Qi Wu [1,2✉]

Contrasting to the established role of the hypothalamic agouti-related protein (AgRP) neurons in feeding regulation, the neural circuit and signaling mechanisms by which they control energy expenditure remains unclear. Here, we report that energy expenditure is regulated by a subgroup of AgRP neurons that send non-collateral projections to neurons within the dorsal lateral part of dorsal raphe nucleus (dlDRN) expressing the melanocortin 4 receptor (MC4R), which in turn innervate nearby serotonergic (5-HT) neurons. Genetic manipulations reveal a bi-directional control of energy expenditure by this circuit without affecting food intake. Fiber photometry and electrophysiological results indicate that the thermo-sensing MC4R$^{dlDRN}$ neurons integrate pre-synaptic AgRP signaling, thereby modulating the post-synaptic serotonergic pathway. Specifically, the MC4R$^{dlDRN}$ signaling elicits profound, bi-directional, regulation of body weight mainly through sympathetic outflow that reprograms mitochondrial bioenergetics within brown and beige fat while feeding remains intact. Together, we suggest that this AgRP neural circuit plays a unique role in persistent control of energy expenditure and body weight, hinting next-generation therapeutic approaches for obesity and metabolic disorders.

[1] USDA/ARS Children's Nutrition Research Center, Baylor College of Medicine, Houston, TX, USA. [2] Department of Pediatrics, Baylor College of Medicine, Houston, TX, USA. [3] Department of Molecular Physiology & Biophysics, Vanderbilt University School of Medicine, Nashville, TN, USA. [4] Department of Neurology and Neurotherapeutics, University of Texas Southwestern Medical Center, Dallas, TX, USA. [5] Pennington Biomedical Research Center, Louisiana State University System, Baton Rouge, LA 70808, United States. [6] Department of Physiology and Biochemistry, Szent Istvan University, Budapeste, Hungary. [7] Department of Comparative Medicine and Department of Neurobiology, Yale University School of Medicine, New Haven, CT, USA. [8]These authors contributed equally: Yong Han, Guobin Xia, Dollada Srisai. ✉email: qiw@bcm.edu

The mammalian central nervous system includes a complex neural network that adaptively regulates feeding and energy expenditure to maintain control over body weight. Evidences suggest that the neural circuits comprised of agouti-related peptide (AgRP) neurons in the arcuate nucleus (ARC), the neural and hormonal signaling afferent to AgRP neurons, as well as their postsynaptic targets, play a pivotal role in governing homeostatic and motivational behaviors, particularly feeding behaviors[1–7]. Rapid manipulation of AgRP neurons or the axonal terminals within several projecting areas can potently regulate food intake and body weight[1,8–12]. We and others found that the AgRP neural circuit regulates feeding behavior by gamma-aminobutyric acid (GABA) signaling; a subset of AgRP neurons promotes feeding, at least partially, by sending inhibitory GABAergic projections to the paraventricular nucleus (PVN) of the hypothalamus and the parabrachial nucleus (PBN) of the brainstem[2,9,13–19]. Recent study also suggests a role of the neuropeptide Y (NPY), which is co-expressed by AgRP neurons for chronic control on appetite[20].

Meanwhile, pharmacological and genetic results indicate that AgRP neurons are involved in the regulation of thermogenesis and energy expenditure[8,21–28]. Chronic administration of AgRP into the third cerebral ventricle decreases oxygen consumption[21,28]. Activation of AgRP neurons induced an obviously decline in thermogenesis and energy expenditure[8,24,25]. In contrast, global deletion of AgRP elevated metabolic rate with age[22]. Genetic inhibition of AgRP neurons activities promoted energy expenditure[26,27,29]. Increasing amount of evidence suggest that melanocortin 4 receptor (MC4R), to which AgRP acts as an inverse agonist, is tightly associated with energy expenditure[30–32]. Some literatures implicated a role of MC4R signaling in brainstem in the control of thermogenesis and energy expenditure[33–37]. However, the neural circuit and signaling pathways underlying the physiological role of AgRP neurons in the control of energy expenditure has not yet characterized.

It has been documented that the brainstem serotonergic system contributes to thermoregulation. Chemogenetic inhibition of serotonergic neurons induced hypothermia[38]. Ablation of central serotonergic neurons disrupts thermoregulation, whereas acute activation a subset of those in the ventral medulla promotes energy expenditure[39,40]. Many evidences show activation of the neural circuit from dorsomedial hypothalamus to the raphe pallidus enhances thermogenesis[41–43], which shows the existence of functional connectivity between the homeostatic systems in the hypothalamus and brainstem.

In this report, we identify an additional link by demonstrating that a subset of caudal AgRP$^{ARC}$ neurons send inhibitory projections to MC4R neurons in the dorsal lateral part of dorsal raphe nucleus (dlDRN), which in turn send glutamatergic projections to a subset of 5-HT neurons in the dorsal medial DRN (dmDRN), thereby establishing a functional neural circuit that controls thermogenesis and energy expenditure without affecting feeding. We show that neuronal ablation and optogenetic/chemogenetic manipulation of AgRP$^{ARC}$ → MC4R$^{dlDRN}$ → 5-HT$^{dmDRN}$ circuit bidirectionally control thermogenesis and energy expenditure, but not food intake. We reveal that suppression of this neural circuit significantly reduces body weight in obese animals through restoration of mitochondrial uncoupling in brown/beige fat tissues.

## Results

**Manipulating AgRP$^{ARC→dlDRN}$ neurons affects energy expenditure not feeding.** Photostimulation of AgRP axon terminals in the dlDRN resulted in a rapid decrease of core body temperature and intrascapular brown adipose tissue (iBAT) temperatures in a frequency-dependent manner (Fig. 1a–e), whereas pretreatment with melanotan II (MTII), an MC3R/MC4R agonist, into the dlDRN completely abolished this effect (Supplementary Fig. 1). To further evaluate the physiological role of AgRP neurons in thermogenesis, the wild-type (WT) mice were freely fed a standard rodent diet or fasted for 24 h. Compared with normal fed states, fasting significantly decreased the core body temperature and iBAT temperature during daytime (Fig. 1d, e). Photostimulation of AgRP terminals in the dlDRN had no effect on feeding in ad libitum-fed mice, whereas activation of the established AgRP → PVN circuit strongly promoted the food intake (Supplementary Fig. 2a, b). To distinguish from other AgRP projections, we found that optogenetic activation of neither the AgRP → PVN feeding circuit nor the AgRP → ventrolateral periaqueductal gray (vlPAG) circuit, which is in close proximity to the dlDRN, exhibits any significant effect on thermogenesis (Supplementary Fig. 2c–e). Then, we functionally labeled AgRP$^{ARC→dlDRN}$ neurons with hM3Dq-mCherry by retrograde labeling the AgRP$^{ARC→dlDRN}$ neurons[44] (Fig. 1f–h and Supplementary Fig. 3a). Systemic application of CNO (1 mg/kg, intraperitoneal (i.p.)) significantly decreased core body temperature and iBAT temperature (Fig. 1i, j), suggesting that manipulating AgRP$^{ARC→dlDRN}$ neurons is sufficient to affect thermogenesis. Furthermore, our results show that AgRP$^{ARC→dlDRN}$ neurons do not send collaterals to other downstream targets by using the same strategy (Supplementary Fig. 3b–j). Histological analysis showed ~1550 AgRP neurons (~17.4%) project to dlDRN (Supplementary Fig. 3k). These data demonstrate that AgRP$^{ARC→dlDRN}$ neurons are sufficient to maintain normal body temperature.

To determine the necessity of AgRP neurons in the regulation of energy expenditure, we used $Agrp^{DTR}$ mice expressing human diphtheria toxin receptors (DTR) exclusively in AgRP neurons, which allows selective and rapid ablation of AgRP neurons upon injecting diphtheria toxin (DT) into brain regions that receive AgRP axon fibers[10,14,45]. The DT concentration in the DRN and nearby regions was checked 24 h after injection of 0.4 ng DT into the dlDRN (Supplementary Fig. 4a). The DT diffusion, to a great extent, was restricted in dlDRN by quantitative calculation based on the absolute liquid chromatography with tandem mass spectrometry (LC–MS) ion intensity values (Supplementary Fig. 4b). These AgRP$^{ARC→dlDRN}$ neurons were ablated by targeted administration of DT into the dlDRN of $Agrp^{DTR/+}::Npy^{GFP}$ mice, where DTR-expressing AgRP neurons were labeled by GFP marker[46] (Fig. 1k). Firstly, we assessed the spatial distribution of AgRP$^{ARC→dlDRN}$ neurons. A progressive deletion of GFP-expressing neurons was observed in the mid-caudal ARC between −1.94 and −2.46 mm bregma on day 4, 7, and 14 after DT injection (Fig. 1l, m and Supplementary Fig. 5). A total of ~1600 AgRP$^{ARC→dlDRN}$ neurons were acutely ablated, which accounts for ~18% of the entire AgRP population (Fig. 1n, o). Then, we examined the effects on thermogenesis. Ablation of AgRP$^{ARC→dlDRN}$ neurons resulted in a significant increase in core body temperature from 36.9 °C (day 1) to 38.0 °C (day 14), and iBAT temperature from 37.6 °C (day 1) to 38.8 °C (day 14; Fig. 1p, q). Ablation of these AgRP neurons increased core body temperature not only from 36.6 °C to 37.8 °C during the daytime, but also from 37.7 °C to 38.6 °C during the nighttime, indicating that manipulating this neural circuit is critical for the regulation of thermogenesis (Supplementary Fig. 6). In contrast to the starvation resulting from ablation of all AgRP neurons or a subset projecting to PBN[10,14], ablation of AgRP$^{ARC→dlDRN}$ neurons resulted in a normal daily food intake (Fig. 1r). A combination of enhanced thermogenesis and intact feeding led to an obvious reduction of body weight over 6 weeks (Fig. 1s). Furthermore, statistical analysis revealed that core body and iBAT temperatures increase linearly with the number of ablated AgRP neurons

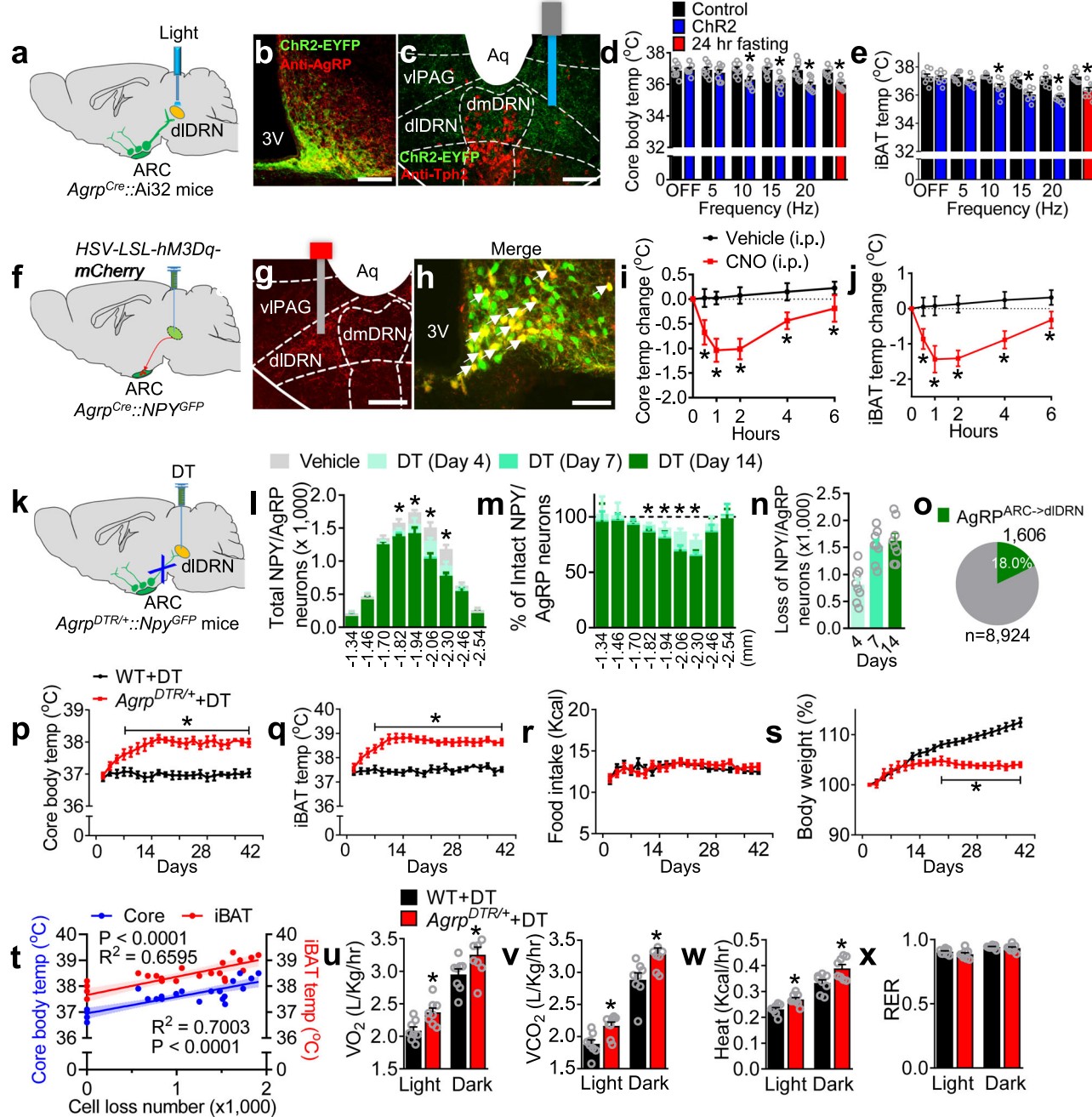

(Fig. 1t). Meanwhile, ablation of AgRP$^{ARC \rightarrow dlDRN}$ neurons led to enhanced $O_2$ consumption, $CO_2$ production, and heat production in both light and dark cycles (Fig. 1u–w). The respiration exchange ratio (RER), however, was not disrupted (Fig. 1x). Although ablation of AgRP neurons transiently activates gliosis in postsynaptic target regions[47], microglia activation in the ARC was not observed 14 days after DT injection into the dlDRN, thereby minimizing the possibility of enhancing thermogenesis due to gliosis effects (Supplementary Fig. 7). In addition, ablation of AgRP$^{ARC \rightarrow dlDRN}$ neurons did not disrupt the integrity of AgRP fibers in the vlPAG or PBN, which are the major hindbrain targets of AgRP neurons (Supplementary Fig. 8).

**Electrophysiological identification of AgRP$^{ARC}$ → MC4R$^{dlDRN}$ → 5-HT$^{dmDRN}$ circuit.** To visually examine the AgRP → dlDRN neural circuit, we labeled the MC4R$^{dlDRN}$ neurons with mCherry. The MC4R$^{dlDRN}$ neurons are in close proximity with AgRP-positive

boutons (Fig. 2a), adjacent but segregated from the nearby 5-HT$^{dmDRN}$ neurons (Fig. 2b and Supplementary Fig. 9). To better understand the connectivity of AgRP neurons and the dlDRN, injection of ZsGreen-tagged wheat germ agglutinin (WGA), a Cre-dependent transsynaptic tracer, allows specific transduction into AgRP neurons[48,49,50] (Supplementary Fig. 10). Again, the results showed that MC4R$^{dlDRN}$ neurons receiving projections from AgRP neurons are distinct from the 5-HT$^{dmDRN}$ neurons. Furthermore, we used the WGA tracer to examine the connectivity of MC4R$^{dlDRN}$ neurons and 5-HT$^{dmDRN}$ neurons (Supplementary Fig. 11a). The tracing data indicated that ~40% of 5-HT$^{dmDRN}$ neurons are innervated by MC4R$^{dlDRN}$ neurons (Fig. 2c, d and Supplementary Fig. 11b–l). Together, these results reveal a neural circuit, in which AgRP neurons in the mid-caudal ARC innervate MC4R$^{dlDRN}$ neurons, which in turn project to a subset of 5-HT$^{dmDRN}$ neurons.

Using this Cre-dependent transsynaptic method, we performed whole-cell, patch-clamp recordings to investigate the synaptic

**Fig. 1 Manipulation of AgRP$^{ARC \to dlDRN}$ neurons bilaterally regulates thermogenesis and energy expenditure without affecting appetite. a** Diagram showing the photostimulation of AgRP axonal terminals in dorsal lateral DRN (dlDRN). **b** Immunostaining image showing the ARC (arcuate nucleus) with merged anti-AgRP (red) and ChR2-EYFP (green). Scale bar, 150 μm. **c** AgRP axonal terminals (green) in the DRN immunostaining with tryptophan hydroxylase 2 (Tph2, red). Scale bar, 100 μm. **d, e** Core body temperature (**d**) and iBAT temperature (**e**) after photostimulation of the AgRP → dlDRN circuit. ($n = 8$ per group; *$P$ was calculated between control and ChR2, or between control and 24 h fasting; for **d**: $F = 6.817$, $P = 0.999$ at OFF, $P = 0.948$ at frequency 5, *$P = 0.0213$ at frequency 10, *$P = 0.0013$ at frequency 15, *$P = 0.0003$ at frequency 20, *$P = 0.0042$ at 24 h fasting; for **e**: $F = 12.35$, $P = 0.998$ at OFF, $P = 0.548$ at frequency 5, *$P = 0.0036$ at frequency 10, *$P < 0.0001$ at frequency 15 and 20, $P = 0.0001$ at 24 h fasting; one-way ANOVA followed by Tukey post hoc test). **f** Schematic image of retrograde labeling of AgRP$^{ARC \to dlDRN}$ neurons by injecting *HSV-hEF1α-LSL-hM3Dq-mCherry* virus into the dlDRN of *Agrp$^{Cre}$::NPY$^{GFP}$* mice. **g** Representative image of virus injection in mice described in **f**. Scale bar, 100 μm. **h** The colocolization of retrograde-labeled AgRP$^{ARC \to dlDRN}$ neurons and NPY-GFP in the ARC. Scale bar, 50 μm. **i, j** Core body temperature (**i**) and iBAT temperature (**j**) changes in mice described in **f**. ($n = 8$ per group; *$P$ was calculated between vehicle and CNO; for **i**: $F = 306.3$, *$P < 0.0001$ at hour 1, 2, 4, and 6; for **j**: $F = 377.1$, *$P < 0.0001$ at hour 1, 2, 4, and 6; two-way ANOVA followed by Bonferroni post hoc test). **k** Diagram showing ablation of a subpopulation of AgRP neurons. **l–n** Quantification of the total number (**l**) and percent (**m**) of NPY/AgRP neurons, and accumulated number of ablated NPY/AgRP neurons (**n**). ($n = 8$ per group; *$P$ was calculated between vehicle and DT (day 14); for **l**, $F = 28.11$, *$P = 0.0105$ at −1.82, *$P < 0.0001$ at −1.94, −2.06, −2.30; for **m**, $F = 6.879$, *$P = 0.027$ at −1.82, *$P = 0.0119$ at −1.94, *$P < 0.0001$ at −2.06, −2.30; two-way ANOVA followed by Bonferroni post hoc test). **o** The pie chart showing the % of AgRP neurons projecting to the dlDRN compared to the total number of AgRP neurons on day 14. **p–s** Core body temperature (**p**), iBAT temperature (**q**), daily food intake (**r**), and body weight (**s**) in mice described in **k**. ($n = 8$ per group; *$P$ was calculated between *Agrp$^{DTR/+}$* + DT and WT + DT; for **p**, $F = 519.4$, *$P < 0.0001$; for **q**, $F = 664.2$, *$P < 0.0001$; for **s**, $F = 300.0$, *$P < 0.0001$; two-way ANOVA followed by Bonferroni post hoc test). **t** Correlation between the number of ablated AgRP$^{ARC \to dlDRN}$ neurons vs core body temperature or iBAT temperature ($n = 6$ per group, linear regression). The shaded areas represent 95% confidence interval for the line of best fit. **u–x** O$_2$ consumption (**u**), CO$_2$ production (**v**), heat production (**w**), and RER (**x**) in mice as described in **k** ($n = 8$ per group; *$P$ was calculated between *Agrp$^{DTR/+}$* + DT and WT + DT; for **u**, $F = 40.19$, *$P = 0.0496$ in light, *$P = 0.0241$ in dark; for **v**, $F = 60.9$, *$P = 0.0457$ in light, *$P = 0.0036$ in dark; for **w**, $F = 40.14$, *$P = 0.0433$ in light, *$P = 0.0023$ in dark; one-way ANOVA followed by Tukey post hoc test). Error bars represent mean ± s.e.m.

connectivity between channelrhodopsin-2 (ChR2)-expressing AgRP axons and postsynaptic neurons in the dlDRN (Fig. 2e). In the presence of CNQX (a competitive AMPA/kainate receptor antagonist), AP5 (a selective NMDA receptor antagonist), and bicuculline (a GABA$_A$ receptor antagonist), photostimulation of the ChR2-expressing AgRP terminals resulted in the robust inhibition of action potential in ~87% of postsynaptic ZsGreen neurons of the DRN in a reversible manner with significant reduction of firing rate (Fig. 2f, h, i). Pretreatment with α-melanocyte-stimulating hormone (α-MSH) (MC3R/MC4R full agonist) abrogated the light-induced suppression (~84%, with α-MSH treatment) on the spontaneous firing of the dlDRN neurons, indicating that presynaptic AgRP signaling directly interacts with melanocortin signaling on the postsynaptic dlDRN neurons (Fig. 2g–i). Furthermore, we labeled the MC4R$^{dlDRN}$ neurons with ChR2 (Fig. 2j). MC4R neurons were confirmed by the response to photostimulation (Fig. 2k). AgRP potently suppressed the firing frequency of all recorded MC4R$^{dlDRN}$ neurons, as a contrast, NPY elicited a moderate suppression on the firing of <50% recorded MC4R$^{dlDRN}$ neurons without affecting thermogenesis (Fig. 2l, m and Supplementary Fig. 12). Moreover, the MC4R$^{dlDRN}$ neurons were depolarized by α-MSH, an endogenous MC4R agonist (Supplementary Fig. 13). Together, these results suggest that AgRP acting on melanocortin signaling accounts for the major signaling route of the AgRP → dlDRN neural circuit.

Synaptic connectivity between MC4R$^{dlDRN}$ and 5-HT$^{dmDRN}$ neurons was further explored electrophysiologically by recording the neural firing of 5-HT$^{dmDRN}$ neurons. The 5-HT neurons innervated by ChR2-expressing MC4R$^{dlDRN}$ neurons were identified by infusing lucifer yellow into cells after recording, followed by immunostaining of anti-tryptophan hydroxylase 2 (Tph2; Fig. 2n). There were ~45% of recorded 5-HT$^{dmDRN}$ neurons receiving projection from MC4R$^{dlDRN}$ neurons (Fig. 2o). To test whether this connectivity was monosynaptic input from MC4R$^{dlDRN}$ axon terminals, we perfused tetrodotoxin (TTX) and 4-aminopyridine (4-AP) into the bath to remove any network activity. We observed that 23.5% of 5-HT$^{dmDRN}$ neurons we tested received direct, monosynaptic excitatory input from MC4R$^{dlDRN}$ axon terminals (Fig. 2p–r). Excitatory postsynaptic currents (EPSCs) in the 5-HT$^{dmDRN}$ neurons triggered by

photostimulation of ChR2-expressing axonal terminals of MC4R$^{dlDRN}$ neurons were fully blocked by the pretreatment with CNQX and AP5 (Fig. 2p–r), confirming that these terminals were releasing glutamate. AgRP exerted strong hyperpolarization of downstream 5-HT$^{dmDRN}$ neurons via an indirect inhibition of the postsynaptic MC4R$^{dlDRN}$ neurons that send glutamatergic afferents onto the 5-HT$^{dmDRN}$ neurons (Supplementary Fig. 14). Moreover, ~60% of recorded 5-HT$^{dmDRN}$ neurons were depolarized by α-MSH, an action that was completely abolished by antagonizing AMPA and/or NMDA receptors (Fig. 2s–w). Together, these data strongly suggest that MC4R$^{dlDRN}$ neurons send glutamatergic projections onto a subset of 5-HT$^{dmDRN}$ neurons via actions upon AMPA (~60%) and NMDA (~40%) glutamate receptor signaling.

**MC4R$^{dlDRN}$ neurons respond to thermogenesis under ambient temperature.** We investigated the importance of MC4R$^{dlDRN}$ neurons in the control of AgRP-mediated thermogenesis under various conditions. Chemogenetic inhibition of MC4R$^{dlDRN}$ neurons decreased core body temperature from 36.8 to 36.0 °C, and iBAT temperature from 37.5 to 36.7 °C under room temperature (RT; Supplementary Fig. 15a, b). It also normalized the AgRP$^{ARC \to dlDRN}$ ablation-induced hyperthermia by reducing core body temperature from 37.9 to 36.6 °C, and iBAT temperature from 38.8 to 37.2 °C (Supplementary Fig. 15a, b). Notably, inactivation of MC4R$^{dlDRN}$ neurons had no effects on food intake (Supplementary Fig. 15c). These results strongly indicate that AgRP ablation-mediated hyperthermia can be reversed upon genetic inhibition of the downstream MC4R$^{dlDRN}$ neurons. We next investigated the calcium activities of MC4R$^{dlDRN}$ neurons under cold challenge. The MC4R$^{dlDRN}$ neurons were labeled with GCaMP6f and hM4Di through virus injection (Supplementary Fig. 16a–f). Fiber photometry analysis in vivo revealed that the calcium transient of MC4R$^{dlDRN}$ neurons were dynamic in response to cold challenge, which was more fluctuating upon AgRP-neuron ablation. Subsequent inhibition of MC4R$^{dlDRN}$ neurons, however, stabilized the calcium transient of MC4R$^{dlDRN}$ neurons (Supplementary Fig. 16g, h). Accordingly, ablation of AgRP$^{ARC \to dlDRN}$ neurons significantly augmented cold-induced thermogenesis, whereas chemogenetic inhibition of MC4R$^{dlDRN}$ neurons severely disrupted this response

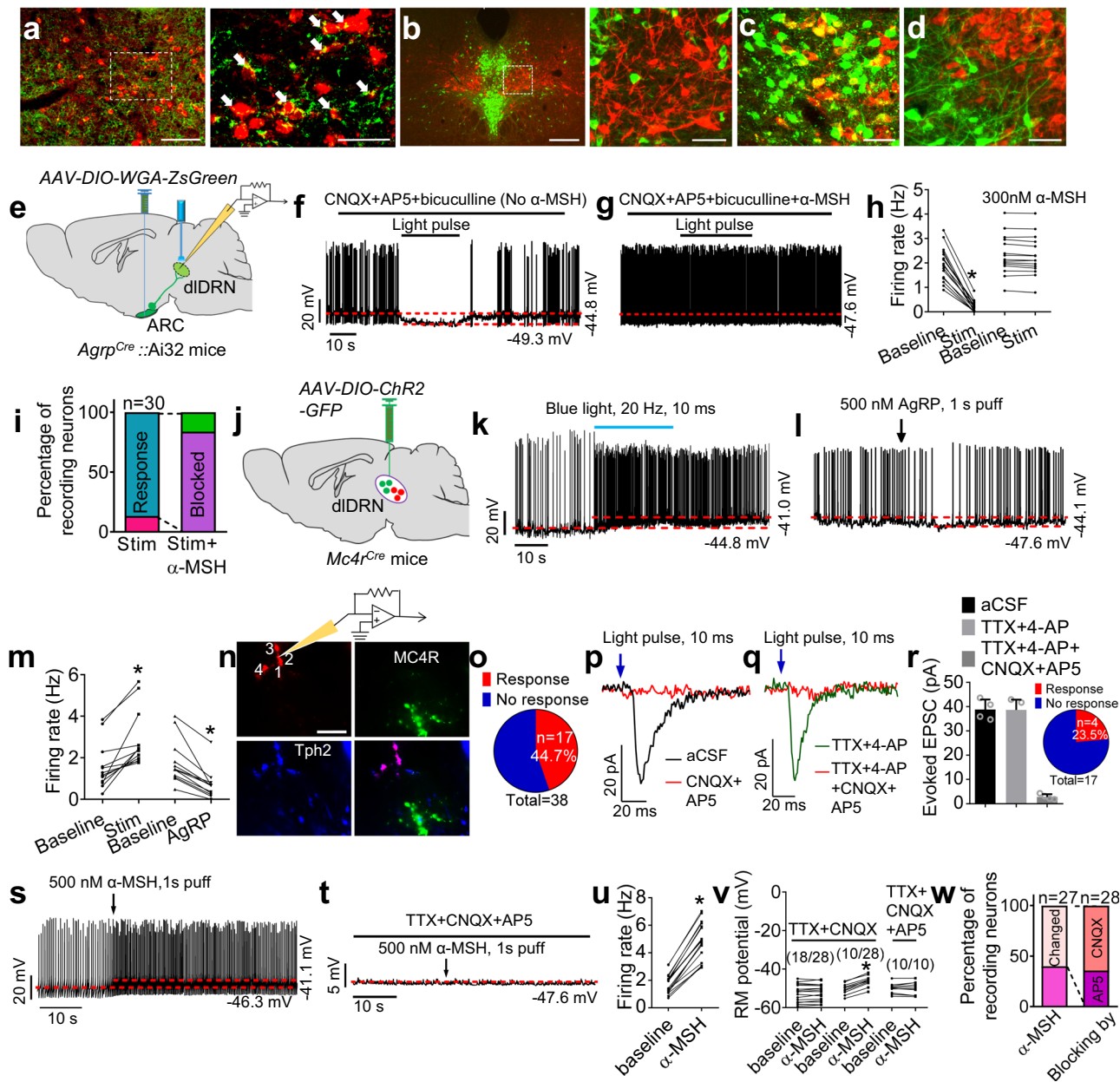

(Supplementary Fig. 16i, j). The fiber photometry signals were unaffected by locomotion (Supplementary Fig. 17). Together, these data indicate that the AgRP[ARC] → MC4R[dlDRN] neural circuit plays a key role in bidirectional control of thermogenesis.

To better understand the electrophysiological connectivity of the AgRP[ARC] → MC4R[dlDRN] → 5-HT[dmDRN] circuit, we employed a combined opto-tetrode recording and cannulation system in vivo in $Agrp^{DTR/+}::Mc4r^{Cre}$ mice treated with $AAV2$-$DIO$-$ChR2$-$GFP$ and $AAV2$-$DIO$-$hM4Di$-$mCherry$ into the dlDRN (Fig. 3a–e). A total 16 MC4R neurons and 10 5-HT neurons were identified through optogenetic-invoked spikes and spontaneous firing rate and pattern[51,52] (Fig. 3f–h and Supplementary Fig. 18). We investigated the activities of MC4R[dlDRN] neurons under thermoneutral (TN) condition within which metabolic heat production (e.g., shivering) or evaporative heat loss (e.g., sweating) are minimized[53]. The results showed that 13 out of 16 identified MC4R neurons responded to a shift from a RT to TN environment, with a reduction of firing rate from 1.35 to 0.87 Hz (Fig. 3i, m, n). We also observed four out of ten putative

5-HT neurons that respond to the ambient temperature changes with a reduction of firing rate from 1.65 to 0.93 Hz (Fig. 3j, m, o). The firing rate of MC4R[dlDRN] neurons was significantly increased to 5.76 Hz after ablation of AgRP neurons under RT (Fig. 3k, m). And the firing rate of putative 5-HT neurons was significantly increased to 5.0 Hz after ablation of AgRP neurons under RT (Fig. 3l, m). We tested that chemogenetic suppression of MC4R neurons in the dlDRN led to a reduction of firing rate of MC4R neurons and putative 5-HT neurons whether the AgRP neurons were ablated or not (Supplementary Fig. 19). As a comparison, vehicle treatment by i.p. had no effects on neural firing of MC4R[dlDRN] neurons and 5-HT neurons (Supplementary Fig. 20). This result indicates that the AgRP → DRN neural circuit mediates thermogenesis predominantly through MC4R and 5-HT signaling.

**Genetic manipulation of MC4R[dDRN] and 5-HT[dmDRN] signaling affects body weight via energy expenditure.** MC4R-null

**Fig. 2 AgRP$^{ARC}$ → dlDRN neurons, MC4R$^{dlDRN}$ neurons, and 5-HT$^{dmDRN}$ neurons organize into a functional neural circuit. a** Representative images showing MC4R$^{dlDRN}$ neurons (red) and AgRP fibers (green) in the dlDRN. *AAV9-DIO-mCherry* was injected into the dlDRN of *Mc4r$^{Cre}$* mice. The arrows indicate AgRP fibers synapsed with MC4R$^{dlDRN}$ neurons. Scale bar in left panel, 100 μm; scale bar in right panel, 50 μm. **b** Representative images showing MC4R$^{dlDRN}$ neurons (red) and 5-HT$^{dmDRN}$ neurons (green). Scale bar in left panel, 200 μm; scale bar in right panel, 50 μm. **c, d** *AAV9-DIO-WGA-ZsGreen* (**c**) or *AAV9-DIO-EGFP* (**d**) injected into the dlDRN of *Mc4r$^{Cre}$* mice, where 5-HT$^{dmDRN}$ neurons (anti-Tph2, red) showed transsynaptic connections from MC4R$^{dlDRN}$ neurons. The yellow neurons in **c** are the 5-HT$^{dmDRN}$ neurons synapsed with MC4R$^{dlDRN}$ neurons. Scale bars in **c** and **d**, 50 μm. **e** Schematic illustration of patch-clamp recording from ZsGreen-expressing dlDRN neurons in *Agrp$^{Cre}$::Ai32* mice with bilateral injection of *AAV9-DIO-WGA-ZsGreen* into the ARC. **f–g** Representative spikes of ZsGreen-labeled dlDRN neurons before and after blue light pulses (10 ms/pulse, 20 Hz) shined onto AgRP axonal fibers (**f**) or with a pretreatment of α-MSH (**g**). **h, i** Firing frequency (**h**) and statistical analysis (**i**) of neurons recorded in **f** and **g** without or with a pretreatment of α-MSH ($n = 30$ neurons from five mice; *$P < 0.0001$; paired two-tailed $t$ test). **j** Schematic illustration showing patch-clamp recordings of MC4R$^{dlDRN}$ neurons or postsynaptic 5-HT$^{dmDRN}$ neurons after injection of *AAV2-DIO-ChR2-GFP* into the dlDRN of *Mc4r$^{Cre}$* mice. **k, l** Representative spikes of MC4R$^{dlDRN}$ neurons with blue light pulses (**k**) or a puff of AgRP (**l**) applied to the mice described in **j**. **m** Firing frequency in MC4R$^{dlDRN}$ neurons of mice described in **k** and **l** ($n = 12$ neurons from three mice; *$P = 0.0004$ in baseline vs stim (stimulation); *$P = 0.0001$ in baseline vs AgRP; paired two-tailed $t$ test). **n** Recording from four identified 5-HT$^{dmDRN}$ neurons innervated by MC4R$^{dlDRN}$ neurons in the mice described in **j**. 5-HT$^{dmDRN}$ neurons were identified by infusing lucifer yellow into cells after recording, followed by immunostaining with anti-Tph2 (blue). Scale bar, 100 μm. **o** Percentage of 5-HT$^{dmDRN}$ neurons identified in all recorded neurons responding or not responding to photostimulation ($n = 38$ neurons from five mice). **p, q** The response of EPSCs recorded from a 5-HT$^{dmDRN}$ neuron upon photostimulation of MC4R$^{dlDRN}$ terminals in dmDRN (10 ms pulse) with or without a pretreatment of CNQX/AP5 (**p**), and in slices pretreated with either TTX + 4-AP or TTX + 4-AP + CNQX + AP5 (**q**). **r** Statistical analysis of light-evoked EPSC amplitudes of mice described in **p** and **q**. Inset showing the total number of 5-HT$^{dmDRN}$ neurons responding or not responding to photostimulation with pretreatment of TTX + 4-AP ($n = 17$ neurons from four mice). **s, t** Representative spikes from 5-HT$^{dmDRN}$ neurons treated with α-MSH (**s**) or α-MSH + TTX + CNQX + AP5 (**t**) in mice described in **n**. **u** Firing frequency of 5-HT$^{dmDRN}$ neurons as showed in **s** ($n = 19$ neurons from four mice; *$P < 0.0001$; paired two-tailed $t$ test). **v** RM (resting membrane) potential of 5-HT$^{dmDRN}$ neurons after application of TTX and CNQX, followed by AP5 ($n = 28$ neurons from four mice; *$P < 0.0001$ in baseline vs α-MSH; paired two-tailed $t$ test). Total 28 5-HT$^{dmDRN}$ neurons were recorded. There are 18 5-HT$^{dmDRN}$ neurons could not be activated by α-MSH, and 10 neurons could be activated by α-MSH in the presence of TTX and CNQX. After application TTX, CNQX, and AP5 these ten neurons failed to respond to α-MSH. **w** Percentage of 5-HT$^{dmDRN}$ neurons depolarized by α-MSH in the presence of CNQX and AP5. Error bars represent mean ± s.e.m.

mice exhibited a reduced basal oxygen consumption[54], displaying defectiveness in acute high fat diet and cold-induced thermogenesis[55]. To assess the physiological role of MC4R$^{dlDRN}$ signaling in the control of energy expenditure, we examined the metabolic response by specific deletion of *Mc4r* in the dlDRN (Supplementary Fig. 21a, b). The results showed that ablation of *Mc4r* in the dlDRN suppressed the energy expenditure and thermogenesis (Fig. 4a–e). These long-term studies showed a reduction of core body temperature but normal body weight during the first 4 weeks (Fig. 4f, h), but during the next 8 weeks there was a progressive increase in body weight, while food intake remained unchanged (Fig. 4g, h). We also genetically reactivated MC4R$^{dlDRN}$ signaling in *Mc4r$^{loxTB}$* mice that carry conditional *Mc4r*-null alleles (Supplementary Fig. 21c, d). Viral-mediated restoration of MC4R$^{dlDRN}$ signaling rescued the hypometabolism phenotypes without affecting the RER (Fig. 4i, l and Supplementary Fig. 21e). We found that restoration of MC4R$^{dlDRN}$ signaling significantly alleviated the obese phenotype in *Mc4r$^{loxTB}$* mice depending upon persistently enhanced energy expenditure and thermogenesis without affecting food intake (Fig. 4m–o). To well establish the chronic role of MC4R$^{dlDRN}$ signaling in AgRP → dlDRN circuit, we genetically disrupted *Mc4r* in the dlDRN of the AgRP$^{ARC}$ → dlDRN neuron-ablated model. This manipulation prevented the progress of hyperthermia caused by the ablation without affecting the food intake (Fig. 4p–r). Together, these results support a critical role of MC4R$^{dlDRN}$ signaling in the bidirectional control of energy expenditure, as well as a significant long-term impact on body weight.

We further investigated the roles of 5-HT$^{dmDRN}$ neurons in thermogenesis and metabolism. Chemogenetic activation or inactivation of 5-HT$^{dmDRN}$ neurons resulted in a significant increase or decrease in iBAT temperature without affecting food intake (Fig. 5a, b and Supplementary Fig. 22). Metabolic results derived from chemogenetic activation or suppression indicate bidirectional control of expenditure parameters by the 5-HT$^{dmDRN}$ neurons (Fig. 5c–e). Notably, a moderate reduction of RER was induced by activation of 5-HT$^{dmDRN}$ neurons, but

that did not account for the enhanced energy expenditure (Fig. 5f). AgRP-induced hypothermia was completely rescued by chemogenetic activation of the 5-HT$^{dmDRN}$ neurons (Fig. 5g). We conclude that a subpopulation of 5-HT$^{dmDRN}$ neurons mediate energy expenditure.

To illuminate the functional hierarchy within the AgRP$^{ARC}$ → MC4R$^{dlDRN}$ → 5-HT$^{dmDRN}$ neural circuit, *AAV9-DIO-WGA-Cre-mCitrine* and *AAV9-Con-Fon-eNpHR-mCherry* virus were injected into the dlDRN and dmDRN, respectively, of *Agrp$^{DTR/+}$::Mc4r$^{Cre}$::Sert$^{Flp}$* mice, allowing specifically control of 5-HT$^{dmDRN}$ neurons that receive projection from MC4R$^{dlDRN}$ neurons in an acute and dynamic manner (Fig. 5h–k). In all postsynaptic dmDRN neurons of MC4R$^{dlDRN}$ neurons, the 5-HT$^{dmDRN}$ neurons receiving projections from MC4R$^{dlDRN}$ neurons accounted for 27.6%. What's more, the 5-HT$^{dmDRN}$ neurons receiving projections from MC4R$^{dlDRN}$ neurons accounted for 36.2% of all 5-HT$^{dmDRN}$ neurons, which specifically targeted the MC4R$^{dlDRN}$ → 5-HT$^{dmDRN}$ circuit (Fig. 5l). We also examined the specificity of viral expression and found 98.4% of 5-HT$^{dmDRN}$ neurons receiving projections from MC4R$^{dlDRN}$ neurons were the postsynaptic neurons of MC4R$^{dlDRN}$ neurons, 97.6% of 5-HT$^{dmDRN}$ neurons receiving projections from MC4R$^{dlDRN}$ neurons were 5-HT neurons (Fig. 5l). The DT was injected into dlDRN to ablate the AgRP$^{ARC}$ → dlDRN neurons, which enhanced the thermogenesis as showed in the previous experiment (Fig. 5m, n). Comparing to several groups that account for critical subset of the control combination, we found that the photoinhibition of 5-HT$^{dmDRN}$ neurons receiving projections from MC4R$^{dlDRN}$ neurons not only suppressed the physiological thermogenesis, but also blunted the enhanced thermogenesis induced by DT ablation of AgRP$^{ARC}$ → dlDRN neurons (Fig. 5m, n). Of note, injection of *AAV9-Con-Fon-eNpHR-mCherry* virus alone had no effects on thermogenesis after photoinhibition (Supplementary Fig. 23a–c). And there was no expression of mCitrine after injection of *AAV9-DIO-WGA-Cre-mCitrine* into the dlDRN with WT mice (Supplementary Fig. 23d). These results demonstrate that AgRP$^{ARC}$ → dlDRN neurons

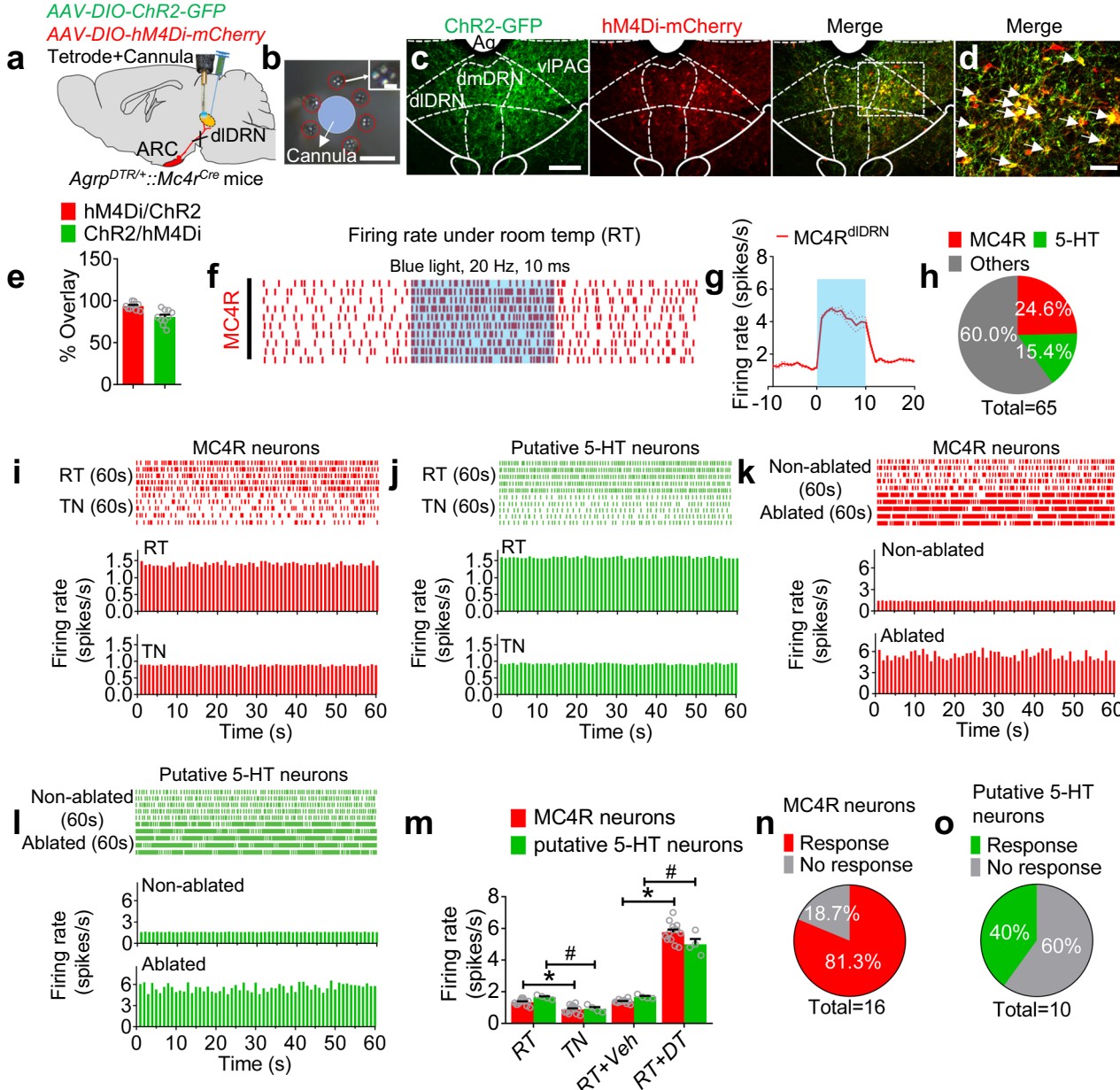

**Fig. 3 The MC4R^dlDRN and 5-HT^dmDRN neurons are fine-tuned by AgRP neurons and ambient temperature. a** Diagram showing in vivo optrode recording in the dlDRN of *Agrp^DTR/+::Mc4r^Cre* mice after injection of *AAV2-DIO-ChR2-GFP* and *AAV2-DIO-hM4Di-mCherry* into the dlDRN. **b** Representative image showing a tetrode bundle consisting of seven tetrodes surrounding a cannula. The inset shows a magnified picture of one single tetrode. Scale bar in **b**, 300 μm; scale bar in inset, 50 μm. **c, d** Representative images showing ChR2-GFP and hM4Di-mCherry in the dlDRN of the mice described in **a**. Scale bar in **c**, 200 μm; scale bar in **d**, 100 μm. **e** Overlap of ChR2 and hM4Di. (*n* = 10 brain sections from four animals per group). **f, g** Raster plots (**f**) and firing rate (**g**) of representative MC4R^dlDRN neurons before, during, and after photostimulation under RT (room temperature; *n* = 16 trials from 16 MC4R^dlDRN neurons). **h** The percentage of MC4R and putative 5-HT neurons out of all recorded neurons in the dlDRN. **i, j** Raster plots (above) and histograms (below) of firing rate of MC4R^dlDRN neurons (**i**) and putative 5-HT^dmDRN neurons (**j**) under RT and TN (thermoneutral) conditions. **k, l** Raster plots and histograms of firing rates of MC4R^dlDRN neurons (**k**) and 5-HT^dmDRN neurons (**l**) under RT conditions with or without AgRP^ARC → dlDRN neurons ablated by DT. **m** Firing rates in MC4R^dlDRN neurons and putative 5-HT^dmDRN neurons under RT, TN conditions, with or without ablation of AgRP^ARC → dlDRN neurons. (*n* = 13 for MC4R^dlDRN neurons, *n* = 4 for 5-HT^dmDRN neurons; *F* = 411.9; for MC4R^dlDRN neurons, *$P$ = 0.0077 RT vs TN, *$P$ < 0.0001 RT + Veh vs RT + DT; for 5-HT^dmDRN neurons, #$P$ = 0.0316 RT vs TN, #$P$ < 0.0001 RT + Veh vs DT + RT; two-way ANOVA followed by Bonferroni post hoc test). **n, o** The percentage of MC4R^dlDRN neurons (**n**) and putative 5-HT^dmDRN neurons (**o**) responding to the RT condition. Error bars represent mean ± s.e.m.

mediate thermogenesis and energy expenditure by signaling to 5-HT^dmDRN neurons via the MC4R^dlDRN neurons.

**The AgRP^ARC → MC4R^dlDRN → 5-HT^dmDRN neural circuit mediates energy expenditure by acting through iBAT and beige WAT.** In the AgRP^ARC → dlDRN neuron-ablated mice model, the

increased core body and iBAT temperatures were fully rescued by SR59230A, a selective β₃ adrenergic receptor (β-3AR) antagonist (Supplementary Fig. 24a). Furthermore, functional blockage of β-3AR completely abolished the increase of core body and iBAT temperatures induced by activation of MC4R^dlDRN neurons or 5-HT^dmDRN neurons (Supplementary Fig. 24b, c). These data

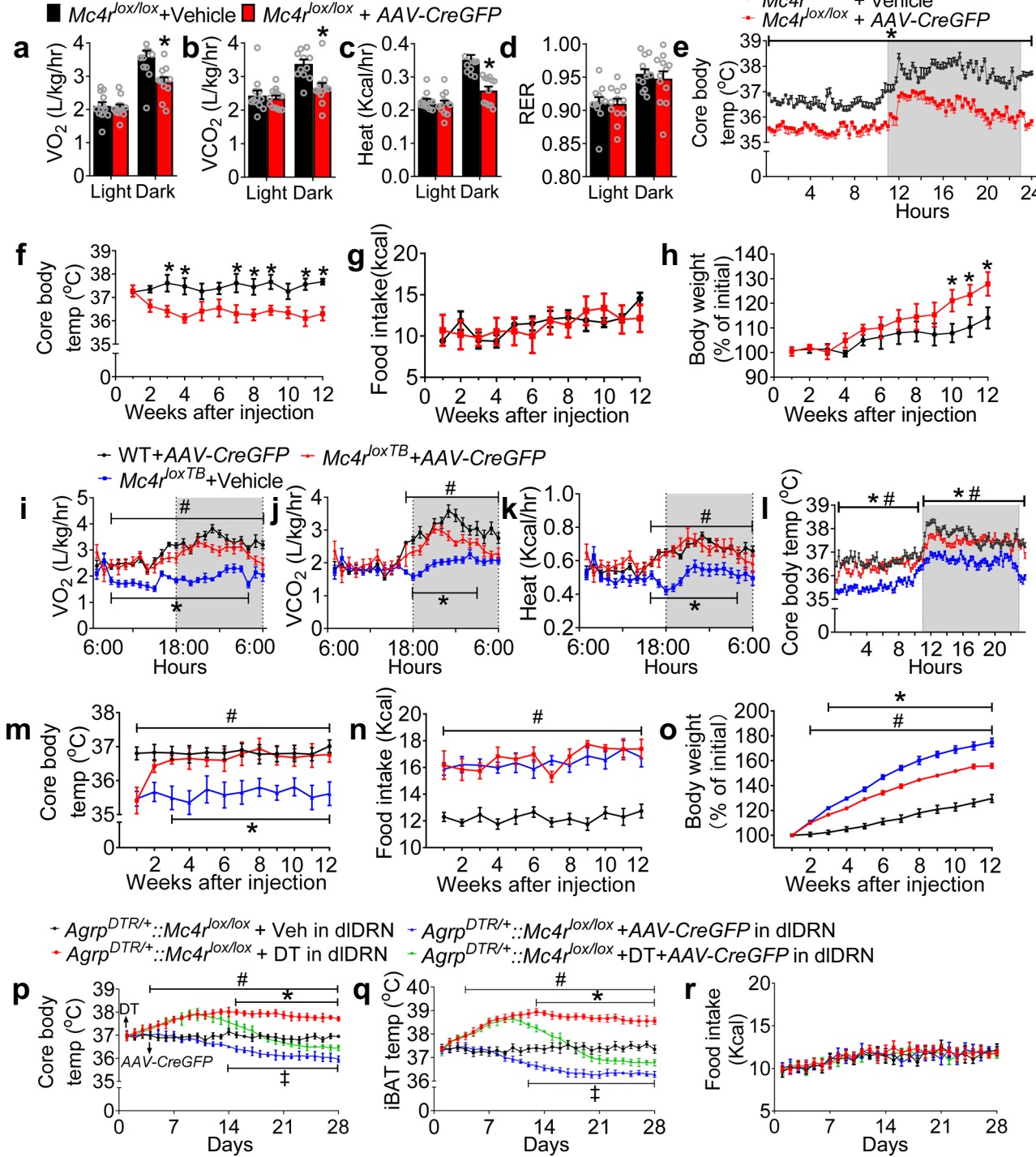

suggest suggesting the sympathetic outflow to the periphery is involved in the thermogenesis mediated by AgRP neural circuits.

To better understand the physiological role of the AgRP^ARC → MC4R^dlDRN → 5-HT^dmDRN circuit in the control of thermogenesis, we examined the cellular and signaling profiles of iBAT and subcutaneous white adipose tissue (scWAT) in response to neural circuit manipulations. We found that knockdown of *Ucp1* in iBAT significantly suppressed the elevated iBAT temperature induced by restoration of MC4R^dlDRN signaling (Fig. 6a–e). We performed a comprehensive morphological, molecular, and cellular analysis on fat tissues before and after restoration of MC4R^dlDRN signaling. Compared to *Mc4r*-null mice and WT

mice, morphological results from the restoration of MC4R^dlDRN signaling revealed an elevated density of mitochondria and smaller lipid droplets in iBAT, as well as clustered scWAT cells with significantly smaller size of adipose cells and beige-like color (Fig. 6f–l). We found that common thermogenesis markers in both scWAT and iBAT were normalized to WT levels after restoration of MC4R^dlDRN signaling (Fig. 6m). More importantly, in comparison to the *Mc4r*-null mice, targeted reexpression of MC4R^dlDRN signaling elicited a significant increase in cellular respiration (Fig. 6n, o). Together, these results reveal that the AgRP^ARC → MC4R^dlDRN → 5-HT^dmDRN neural circuit bidirectionally modulates thermogenesis by targeting the mitochondrial

**Fig. 4 MC4RdlDRN signaling regulates energy expenditure without affecting food intake. a–d** $O_2$ consumption (**a**), $CO_2$ production (**b**), heat production (**c**), and RER (**d**) of $Mc4r^{lox/lox}$ mice injected with vehicle ($AAV2$-$GFP$) or $AAV2$-$CreGFP$ in the dlDRN. ($n = 12$ per group; for **a**: $F = 8.263$, $*P = 0.008$; for **b**: $F = 9.656$, $*P = 0.006$; for **c**: $F = 20.98$, $*P < 0.0001$; two-way ANOVA followed by Bonferroni post hoc test). **e** Core body temperature monitored for 24 h in mice described in **a–d** ($n = 8$ per group; $F = 1790$, $*P < 0.0001$; two-way ANOVA followed by Bonferroni post hoc test). **f–h** Core body temperature (**f**), food intake (**g**), and body weight (**h**) for 12 weeks in the mice described in **a–d** ($n = 8$ per group; for **f**: $F = 76.13$, $*P = 0.046$ at week 3, $*P = 0.0118$ at week 4, $*P = 0.0196$ at week 7, $*P = 0.046$ at week 8, $*P = 0.042$ at week 9, $*P = 0.0057$ at week 11, $*P = 0.0118$ at week 12; for **h**: $F = 130.2$, $*P < 0.0001$ at week 10, 11, 12; two-way ANOVA followed by Bonferroni post hoc test). **i–l** $O_2$ consumption (**i**), $CO_2$ production (**j**), heat production (**k**), and core body temperature for a 24-h period (**l**) 2 weeks after injection of vehicle ($AAV2$-$GFP$) or $AAV2$-$CreGFP$ in the dlDRN of $Mc4r^{loxTB}$ or WT mice. ($n = 8$ per group; $*P$ was calculated between $Mc4r^{loxTB}$ + vehicle and $Mc4r^{loxTB}$ + $AAV2$-$CreGFP$, $\#P$ was calculated between $Mc4r^{loxTB}$ + vehicle and WT + $AAV2$-$CreGFP$; for **i**: $F = 272.4$, $*P < 0.0001$, $\#P < 0.0001$; for **j**: $F = 96.88$, $*P < 0.0001$, $\#P < 0.0001$; for **k**: $F = 100.8$, $*P < 0.0001$, $\#P < 0.0001$; for **l**: $F = 850.2$, $*P < 0.0001$, $\#P < 0.0001$; two-way ANOVA followed by Bonferroni post hoc test). **m–o** Core body temperature (**m**), food intake (**n**), and body weight (**o**) in mice described in **i–l** for 12 weeks after viral injection ($n = 8$ per group; $*P$ was calculated between $Mc4r^{loxTB}$ + vehicle and $Mc4r^{loxTB}$ + $AAV2$-$CreGFP$, $\#P$ was calculated between $Mc4r^{loxTB}$ + vehicle and WT + $AAV2$-$CreGFP$; for **m**: $F = 64.94$, $*P < 0.0001$, $\#P < 0.0001$; for **n**: $F = 235.1$, $*P < 0.0001$, $\#P < 0.0001$; for **o**: $F = 688.2$, $*P < 0.0001$, $\#P < 0.0001$; two-way ANOVA followed by Bonferroni post hoc test). **p–r** Core body temperature (**p**), iBAT temperature (**q**), and food intake (**r**) for 28 days after injection of $AAV2$-$GFP$ (vehicle), DT, $AAV2$-$CreGFP$ or DT + $AAV2$-$CreGFP$ into the dlDRN of $Agrp^{DTR/+}::Mc4r^{lox/lox}$ mice ($n = 8$ per group; $*P$ was calculated between DT and DT + $AAV2$-$CreGFP$, $\#P$ was calculated between DT and vehicle; $\ddagger P$ was calculated between $AAV2$-$CreGFP$ and vehicle; for **p**: $F = 337.9$, $*P < 0.0001$, $\#P < 0.0001$, $\ddagger P < 0.0001$; for **q**: $F = 741.1$, $*P < 0.0001$, $\#P < 0.0001$, $\ddagger P < 0.0001$; two-way ANOVA followed by Bonferroni post hoc test). Error bars represent mean ± s.e.m.

machinery of the peripheral iBAT and beige scWAT in both lean and obese subjects. The studies in this paper establish a central $AgRP^{ARC} \rightarrow MC4R^{dlDRN} \rightarrow 5\text{-}HT^{dmDRN}$ neural circuit with mechanistic significance that controls thermogenesis and energy expenditure bidirectionally and can effectively reverse obesity (Fig. 6p).

## Discussion

Increasing literatures suggest that the DRN plays a role in the control of thermogenesis[56–59]. Here, we delineate an $AgRP^{ARC} \rightarrow MC4R^{dlDRN} \rightarrow 5\text{-}HT^{dmDRN}$ neural circuit, in which a subpopulation of AgRP neurons send axonal projections to the MC4R neurons in the lateral dlDRN, which subsequently innervate a subset of $5\text{-}HT^{dmDRN}$ neurons to mediate thermogenesis via sympathetic outflow to iBAT and beige scWAT (Fig. 6p). Moreover, we show that AgRP, MC4R, glutamate, and serotonin account for the key neurotransmitter signaling molecules in this circuit. Acute ablation of $AgRP^{ARC \rightarrow dlDRN}$ neurons results in disinhibition of $MC4R^{dlDRN}$ neurons that subsequently activates a subset of $5\text{-}HT^{dmDRN}$ neurons, thereby resulting in potent elevation of thermogenesis. The enhanced metabolic effects caused by ablation of $AgRP^{ARC}$ neurons that project to the DRN can be efficiently abolished by optogenetic suppression of $5\text{-}HT^{dmDRN}$ neurons receiving projections from $MC4R^{dlDRN}$ neurons. Together, these findings reveal that a subpopulation of AgRP neurons specifically mediates energy expenditure by suppressing serotonergic signaling in the dlDRN via a MC4R-dependent mechanism.

Previous studies have shown that MC4R signaling located within the brainstem and sympathetic system mediates diet-induced thermogenesis[35,55]. In this study, genetic manipulation of the mid-brain $MC4R^{dlDRN}$ signaling downstream of $AgRP^{ARC}$ neurons elicits bidirectional control of thermogenesis and energy expenditure. Genetic reactivation of $MC4R^{dmDRN}$ signaling increases thermogenesis and energy expenditure without affecting the food intake. In contrast, deletion of $MC4R^{dlDRN}$ signaling not only suppresses metabolic rate, but also causes a significant reduction of thermogenesis. The most profound changes are that restoration of $MC4R^{dlDRN}$ signaling in an obese model results in a late-onset decrease in body weight due to the prolonged change of energy metabolism. Four weeks after restoring $MC4R^{dlDRN}$ signaling on $Mc4r$-null background, there was a significant reduction of body weight as a result of normalized metabolic rates while the hyperphagic phenotype contributed by MC4R deficiency was still observed. We observed a 12% reduction of body weight 12 weeks after restoring $MC4R^{dlDRN}$ signaling. This hypermetabolism-induced weight loss is considerable as comparable to that is achieved by restoration of the MC4R signaling within the PVN that affects feeding[60]. Conversely, mice with deletion of $MC4R^{dlDRN}$ signaling showed no effects on feeding after 3 months, but maintain hypometabolism, ultimately leading to a significant body weight gain. These studies affirm that the $AgRP^{ARC} \rightarrow MC4R^{dlDRN}$ neural circuit elicits profound bidirectional effects on body weight control via strict regulation of energy expenditure.

Our findings further demonstrate that a subset of $5\text{-}HT^{dmDRN}$ neurons play a critical role in the control of thermogenesis. The DRN is known to be a heterogeneous structure in which different subpopulations of intermingled 5-HT neurons may control distinct physiological and psychiatric processes[61]. For example, a recent report shows that a subpopulation of Vglut3-expressing 5-HT neurons within the ventral medial DRN exclusively mediates appetite[62]. We further prove these glutamatergic, non-serotonergic, $MC4R^{dlDRN}$ neurons promote thermogenesis, which is exactly opposite to the role of the neighboring $GABA^{DRN}$ neurons on suppression of energy expenditure[59]. What's more, functionally inactivating $5\text{-}HT^{dmDRN}$ neurons not only suppressed thermogenesis under physiological condition, but also decrease thermogenesis to the equal extent, while $AgRP^{ARC \rightarrow dlDRN}$ neurons were ablated. Future study that aims to characterize those $5\text{-}HT^{dmDRN}$ neurons involved in thermogenesis via comprehensive transcriptome profiling would facilitate the discovery of more critical genes and signaling cascades in the control of body weight and energy homeostasis.

The 5-HT neurons in the DRN display diverse in vivo spiking behaviors[63–66]. One subgroup of 5-HT neurons could be identified by electrophysiological characteristics, including firing rate, firing rhythmicity, and spike distribution, which displayed slow-firing clock-like pattern[52,67,68]. In our studies, the putative 5-HT neurons showed the low frequency (1.65 Hz) with a highly regular pattern that was revealed by the narrow interspike interval (Fig. 3j, m). Moreover, 5-HT neurons displayed a pacemaker pattern where autocorrelation histograms typically exhibited two or three regular peaks. All of characteristics indicated that these recorded cells are 5-HT neurons. However, classical electro-physiological identification criteria may misidentify a subpopulation of non-5-HT neurons[52,68]. In our studies, there were 40% putative 5-HT neurons responding to the switch from RT to TN condition (Fig. 3j, m, o), suggesting that false-positive 5-HT neurons may be inadvertently included by the

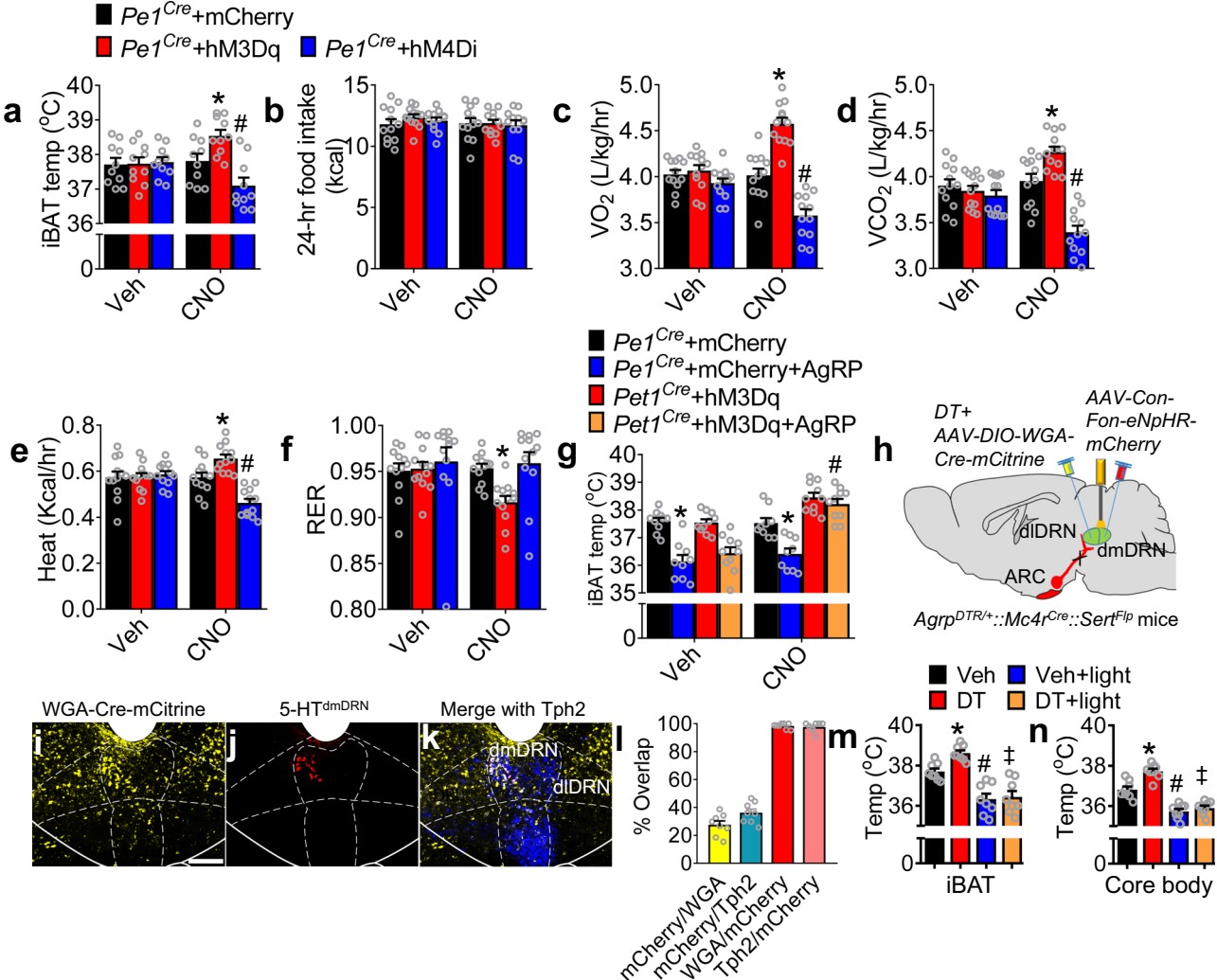

**Fig. 5 5-HT^dmDRN signaling regulates energy expenditure without affecting food intake. a** iBAT temperature measured 2 h after i.p. injection of vehicle (0.9% NaCl) or CNO (1 mg/kg, i.p.) in *Pet1^Cre* mice with *AAV2-DIO-hM3Dq-mCherry* or *AAV2-DIO-hM4Di-mCherry* into the dmDRN. **b** Depiction of 24-h food intake of mice described in **a**, measured on day 7 after CNO injection (twice per day, 1 mg/kg, i.p.). **c–f** $O_2$ consumption (**c**), $CO_2$ production (**d**), heat production (**e**), and RER (**f**) after administration of vehicle (0.9% NaCl) or CNO (twice per day, 1 mg/kg, i.p.) in mice described in **a** ($n = 10$ in **a** per group, $n = 12$ in **b–f** per group; *$P$ was calculated between *Pet1^Cre* + mCherry and *Pet1^Cre* + hM3Dq, #$P$ was calculated between *Pet1^Cre* + mCherry and *Pet1^Cre* + hM4Di; for **a**: $F = 6.711$, *$P = 0.0281$, #$P = 0.0341$; for **c**: $F = 41.76$, *$P < 0.0001$, #$P < 0.0001$; for **d**: $F = 26.28$, *$P = 0.0035$, #$P < 0.0001$; for **e**: $F = 11.49$, *$P = 0.01$, #$P = 0.0003$; for **f**: $F = 3.535$, *$P = 0.026$; two-way ANOVA followed by Bonferroni post hoc test). **g** iBAT temperatures in *Pet1^Cre* mice (by injection of vehicle or *AAV2-DIO-hM3Dq-mCherry* into dmDRN) after microinjection of either vehicle (0.9% NaCl) or AgRP into dlDRN with i.p. treatment with vehicle or CNO ($n = 9$ per group; *$P$ was calculated between *Pet1^Cre* + mCherry and *Pet1^Cre* + mCherry + AgRP, #$P$ was calculated between *Pet1^Cre* + hM3Dq and *Pet1^Cre* + hM3Dq + AgRP; $F = 32.5$, for Veh, *$P < 0.0001$; for CNO, *$P = 0.003$, #$P > 0.05$; two-way ANOVA followed by Bonferroni post hoc test). **h** Diagram showing viral targeting of the AgRP^ARC → MC4R^dlDRN → 5-HT^dmDRN circuit by injection of *AAV9-DIO-WGA-Cre-mCitrine* into the dlDRN and *AAV9-Con-Fon-eNpHR-mCherry* into the dmDRN. DT was injected into dlDRN 4 weeks after virus injection with an optical fiber inserted into the dmDRN of *Agrp^DTR/+::Mc4r^Cre::Sert^Flp* mice. The iBAT and core body temperature were measured 7 days after DT injection. **i–k** The transsynaptic labeling by WGA-Cre-mCitrine in the DRN (**i**), specifically labeling of 5-HT^dmDRN neurons receiving projections from MC4R^dlDRN neurons by eNpHR-mCherry (**j**), and immunostaining of anti-Tph2 (**k**). Scale bar in **i** for **i–k**, 100 μm. **l** Statistical data showing the percentage of 5-HT^dmDRN neurons receiving projections from MC4R^dlDRN neurons in all postsynaptic neurons of MC4R^dlDRN neurons in dmDRN and all 5-HT^dmDRN neurons, as revealed in mice described in **h. m, n** The iBAT temperature (**m**) and core body temperature (**n**) 1 h after photoinhibition in the mice described in **h** ($n = 8$ per group; *$P$ was calculated between vehicle and DT, #$P$ was calculated between vehicle and vehicle + light, ‡$P$ was calculated between DT and DT + light; for **m**: $F = 23.82$, *$P = 0.0353$, #$P = 0.0009$, ‡$P < 0.0001$; for **n**: $F = 48.63$, *$P = 0.0002$, #$P < 0.0001$, ‡$P < 0.0001$; one-way ANOVA followed by Tukey post hoc test). Error bars represent mean ± s.e.m.

electrophysiological recording method. A combination of optogenetic and in vivo electrophysiological recording approaches may minimize this problem and identify a cohort of 5-HT neurons[69,70].

Genetic perturbations within this AgRP^ARC → MC4R^dlDRN → 5-HT^dmDRN circuit elicit profound effects on the expression profile of key thermogenesis markers in a way that is consistent

with the behavioral and metabolic phenotypes. Moreover, the genetic enhancement of MC4R^dlDRN signaling results in hypermetabolism of both iBAT and beige scWAT, as demonstrated by tissue morphology, profiling of thermogenesis markers, and mitochondrial respiration rates. Evidence from the neural tracing, as well as pharmacological study also suggests that this neural circuit depends upon the sympathetic outflow to mediate energy

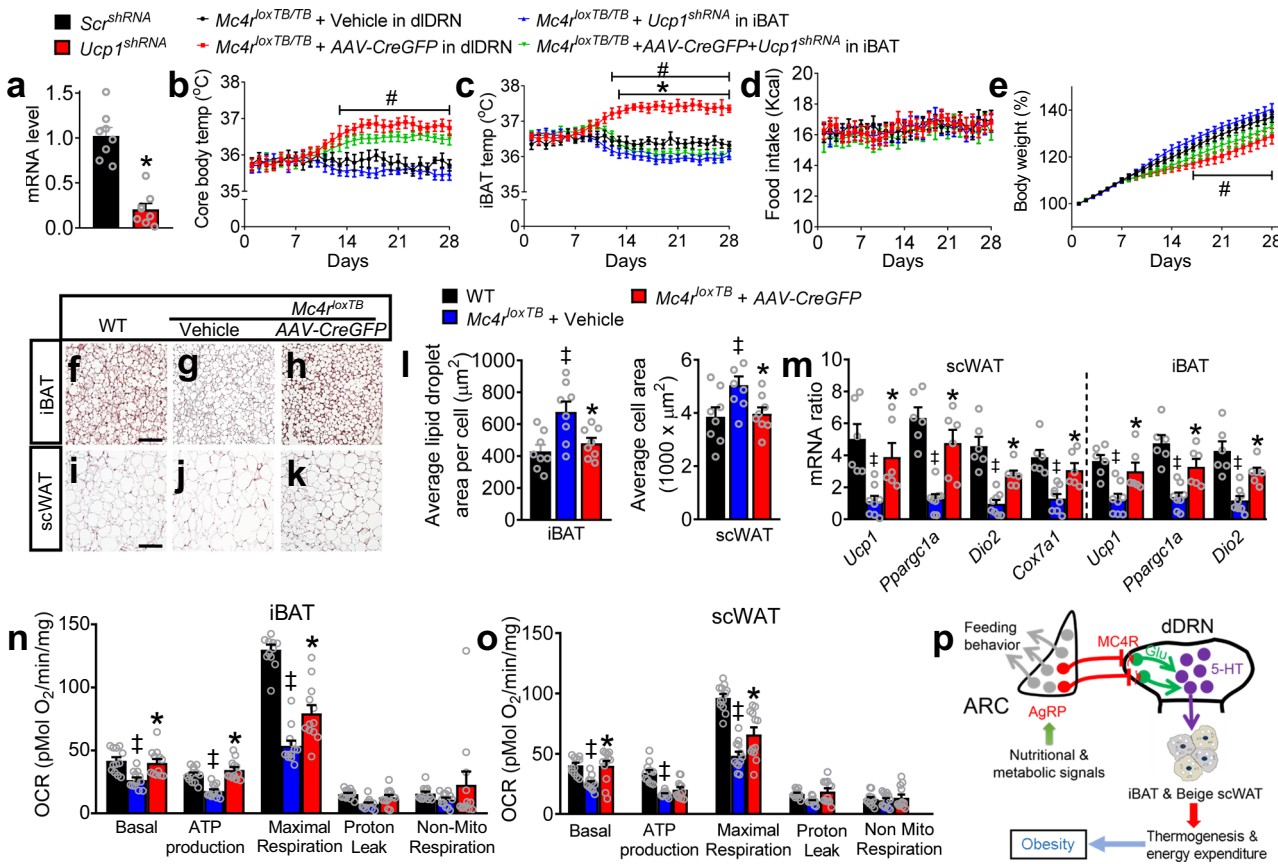

**Fig. 6 The AgRP^ARC → MC4R^dlDRN → 5-HT^dmDRN neural circuit mediates thermogenesis through brown and beige adipose tissues. a** The expression profile of *Ucp1* in the iBAT was analyzed by qRT-PCR 4 weeks after injection of the lentivirus expressing *UCP1^shRNA* or control *SCR^shRNA* into the iBAT ($n = 8$ per group; $F = 2.160$, *$P < 0.0001$; unpaired two-tailed *t* test). **b–e** Core body temperature (**b**), iBAT temperature (**c**), food intake (**d**), and body weight (**e**) monitored for 4 weeks in the mice described in **a** ($n = 8$ per group; *$P$ was calculated between *AAV2-CreGFP* and *AAV2-CreGFP* + *Ucp1^shRNA*, #$P$ was calculated between vehicle and *AAV2-CreGFP*; for **b**, $F = 130.0$, #$P < 0.0001$; for **c**, $F = 264.8$, *$P < 0.0001$, #$P < 0.0001$; for **e**, $F = 88.89$, #$P = 0.0002$; two-way ANOVA followed by Bonferroni post hoc test). **f–k** Representative HE staining of the iBAT (**f–h**) and scWAT (**i–k**) from 14- to 16-week-old WT, *Mc4r^loxTB* mice with vehicle (*AAV2-GFP*) or *AAV2-CreGFP* injected into the dlDRN. Scale bar in **f** and **i** for **f–h** and **i–k**, 50 μm. **l** Average lipid droplet area in iBAT and adipocyte area in scWAT in mice described in **f–k** ($n = 8$ per group; *$P$ was calculated between *Mc4r^loxTB* + vehicle and *Mc4r^loxTB* + *AAV2-CreGFP*, ‡$P$ was calculated between *Mc4r^loxTB* + vehicle and WT; for left panel: $F = 7.218$, *$P = 0.0277$, ‡$P = 0051$; for right panel: $F = 2.943$, *$P = 0·0472$, ‡$P = 00274$; one-way ANOVA followed by Tukey post hoc test). **m** qRT-PCR profile of key thermogenesis markers in the scWAT and iBAT of the mice described in **f–k** ($n = 6$ in WT and *Mc4r^loxTB* + *AAV2-CreGFP* group, $n = 8$ in *Mc4r^loxTB* + vehicle group; *$P$ was calculated between *Mc4r^loxTB* + vehicle and *Mc4r^loxTB* + *AAV2-CreGFP*, ‡$P$ was calculated between *Mc4r^loxTB* + vehicle and WT; for scWAT, $F = 56.34$, *$P = 0.0010$, ‡$P < 0.0001$ in *Ucp1*, *$P < 0.0001$, ‡$P < 0.0001$ in *Ppargc1a*, *$P = 0.0445$, ‡$P < 0.0001$ in *Dio2*, *$P = 0.0468$ ‡$P = 0.0022$ in *Cox7a1*; for iBAT, $F = 41.69$, *$P = 0.0095$, ‡$P = 0.0003$ in *Ucp1*, *$P = 0.0049$, ‡$P < 0.0001$ in *Ppargc1a*, *$P = 0.0097$, ‡$P < 0.0001$ in *Dio2*; two-way ANOVA followed by Bonferroni post hoc test). **n**, **o** Oxygen consumption rate (OCR) for basal respiration, ATP production, maximal respiration, proton leak, and nonmitochondrial respiration are measured in the mice described in **f–k**. ($n = 12$ per group; *$P$ was calculated between *Mc4r^loxTB* + vehicle and *Mc4r^loxTB* + *AAV2-CreGFP*, ‡$P$ was calculated between *Mc4r^loxTB* + vehicle and WT; for **n**: $F = 46.56$, *$P = 0.0392$, ‡$P = 0.0165$ in basal, *$P = 0·007$, ‡$P = 0.0481$ in ATP production, *$P < 0.0001$, ‡$P < 0.0001$ in maximal respiration; for **o**, $F = 53.72$, *$P = 0.0005$, ‡$P = 0.0003$ in basal, ‡$P < 0.0001$ in ATP production, *$P < 0·0001$, ‡$P < 0.0001$ in maximal respiration; two-way ANOVA followed by Bonferroni post hoc test). **p** A schematic diagram showing a neural circuit, in which a subset of AgRP^ARC neurons send inhibitory projections to MC4R^dlDRN neurons, which in turn send glutamatergic projections onto postsynaptic 5-HT^dmDRN neurons. This neural circuit orchestrates bidirectional control of energy expenditure with an enduring and profound effect against body weight in obesity. Error bars represent mean ± s.e.m.

expenditure. These results reveal that the AgRP → dlDRN neural circuit segregates the control of energy expenditure from other established feeding circuits. Our work further suggests a therapeutic potential by targeting central thermoregulatory neural circuits in combating obesity and other metabolic complications.

## Methods

**Animal.** All animal care and experimental procedures were approved by the Institutional Animal Care and Use Committees at Baylor College of Medicine and were performed in accordance with the guidelines described in the NIH guide for the care and use of laboratory animals. Mice used for data collection were at least 8-week-old males; and kept in a temperature (22 °C) and humidity (40–60%)-controlled rooms,

in a 12/12 h light/dark cycle, with lights on from 6:00 a.m. to 6:00 p.m. At the time of the experiments, animals were 8–16 weeks old. Health status was normal for all animals. *Agrp^DTR* mice[10], *Npy^GFP* mice[46], *Agrp^Cre* mice[13], *Mc4r^Cre* mice[3], *Sert^Flp* mice[71], *Mc4r^loxTB* mice[37], *Mc4r^lox/lox* mice[72], *Pet1^Cre* mice[73], Ai32 mice[74], and Ai14 (or *Rosa26^tdTomato*) mice[74] were produced, as previously described. All mice are on a C57Bl/6 background with at least eight generations backcrossed.

**General surgical procedures.** All the mice with brain surgery were performed with the same preoperative and postoperative care. In all cases, preoperative analgesia: 0.1 mg/kg buprenorphine (s.c., 1 h prior to the start of anesthesia). Animals were anesthetized with ~2% isoflurane anesthesia and placed on a stereotaxic frame (David Kopf, Tujunga, CA). A line block of local anesthetics (50/50 mix of lidocaine and bupivicaine in 1:20 dilution; 0.1 ml per 25 g mouse) was made

before making an incision. Postoperative analgesia: 0.1 mg/kg buprenorphine (s.c.) will be administered for once 72 h after surgery. All surgeries were performed having the animals placed on a heating pad and allowed to recover in a heating cage until they chose to reside in the unheated side of the cage.

**Stereotaxic viral injections.** After anesthetization, a circular craniotomy was drilled at the location per different experiments. In all experiments, virus was loaded into a needle (Hamilton Small Hub RN 33 G, Reno, NV) connected with a 10 µl syringe (Hamilton 700 Microliter, Reno, NV). Injections were performed with an Ultra MicroPump (World Precision Instruments, Sarasota, FL) and Micro4 Controller (Heidenhain Corporation, Schaumburg, IL), at a rate of 0.1 µl/min. For AAV, a total 0.5 µl volume was delivered into brain regions. For HSV, a total 0.2 µl was delivered into the dlDRN. The relevant stereotaxic coordinates for the injections are described in the following according to a standardized atlas of the mouse brain (Franklin and Paxinos, third edition, 2007). To target the ARC, the coordinate is $AP = -2.06$ mm, $ML = \pm 0.25$ mm, $DV = -5.9$ mm, the viral aliquots *AAV9-DIO-WGA-zsGreen* (packaged by Optogenetics and Viral Design/Expression Core at Baylor College of Medicine and diluted to a final working titer of $3.8 \times 10^{12}$ viral genomes per ml) were injected. To target the dlDRN the coordinate is $AP = -4.36$ mm, $ML = \pm 0.35$ mm, $DV = -3.3$ mm, the viral aliquots (*AAV9-DIO-mCherry* and *AAV9-DIO-EGFP* packaged by Optogenetics and Viral Design/Expression Core at Baylor College of Medicine and diluted to a final working titer of $6 \times 10^{12}$ viral genomes per ml; *AAV2-EF1a-DIO-hChR2(E123T/T159C)-GFP* from UNC and diluted to a final working titer of $7 \times 10^{12}$ viral genomes per ml; *AAV2-GFP* and *AAV2-CreGFP* packaged by Optogenetics and Viral Design/Expression Core at Baylor College of Medicine and diluted to a final working titer of $5 \times 10^{12}$ viral genomes per ml; *AAV2-hSyn-DIO-hM4D(Gi)-mCherry*; *AAV5-CAG-Flex-GCaMP6f* from UNC and diluted to a final working titer of $7 \times 10^{12}$ viral genomes per ml; *AAV9-DIO-WGA-zsGreen*; *AAV9-DIO-WGA-Cre-mCitrine* synthesized by GenScrip, Piscataway, NJ, USA and packaged by Optogenetics and Viral Design/Expression Core at Baylor College of Medicine and diluted to a final working titer of $3.8 \times 10^{12}$ viral genomes per ml; *HSV-hEF1α-LSL-hM3D(Gq)-mCherry* from Harvard Gene Delivery Technology Core and diluted to a final working titer of $8 \times 10^{12}$ viral genomes per ml) were respectively injected. To target the dmDRN, the coordinate is $AP = -4.60$ mm, $ML = \pm 0$ mm, $DV = -3.1$ mm, the viral aliquots (*AAV2-hSyn-DIO-hM3D(Gq)-mCherry*; *AAV2-hSyn-DIO-hM4D (Gi)-mCherry*; *AAV2-GFP*; *AAV2-CreGFP*; *AAV9-DIO-eNpHR-mCherry* from UNC and diluted to a final working titer of $6 \times 10^{12}$ viral genomes per ml; *AAV9-Con-Fon-eNpHR-mCherry* packaged by Optogenetics and Viral Design/Expression Core at Baylor College of Medicine and diluted to a final working titer of $7 \times 10^{12}$ viral genomes per ml) were respectively injected.

**Ablation of AgRP neurons.** To ablate AgRP neurons, mice carrying the $Agrp^{DTR/+}$ allele, were bilaterally microinjected with DT (0.4 ng/side/mouse, List Biological Laboratories, Campbell, CA) using a needle (Hamilton Small Hub RN 33 G, Reno, NV) connected with a 10 µl syringe (Hamilton 700 Microliter, Reno, NV), at a rate of 0.1 µl/min. A total 0.4 µl volume/side was delivered into the dlDRN[2]. To describe the expression profile of $AgRP^{ARC-dlDRN}$ neurons, the vehicle (0.9% NaCl) or DT were injected into the dlDRN with $Agrp^{DTR/+}::Npy^{GFP}$ mice. The 0, 4, 7, and 14 days after injection, the mice were euthanized and the brain was harvested. The whole ARC was sectioned with 20 µm thickness by a microtome (ThermoFisher Scientific, Waltham, MA). Fluorescent images of GFP-labeled NPY/AgRP neurons along the rostral to caudal axis of the ARC were obtained by an Axio Observer microscope (Zeiss, Thornwood, NY) and further analyzed using ImageJ software version 1.49 (NIH), in which Abercrombie's correction of overcounting of profiles in sections was applied[75].

**Liquid chromatography with tandem mass spectrometry.** To evaluate the effects of DT diffusion in the DRN and surrounding brain regions, the relative score for DT concentration by LC–MS was calculated 24 h after DT injection into the dlDRN (Supplementary Fig. 4a). The LC–MS was performed by the Clinical and Translational Proteomics Service Center at the University of Texas Health Science Center. The brain tissues were punched 24 h after DT (0.4 ng) injection into the dlDRN. Four punches containing the dlDRN ($AP = -4.36$ mm, $ML = \pm 0.35$ mm, $DV = -3.3$ mm), vlPAG ($AP = -4.60$ mm, $ML = \pm 0.50$ mm, $DV = -2.8$ mm), dmDRN ($AP = -4.60$ mm, $ML = \pm 0$ mm, $DV = -3.1$ mm), and mRT ($AP = -4.36$ mm, $ML = \pm 0.75$ mm, $DV = -3.5$ mm) were collected with the thickness of 1.0 mm by brain matrix. The brain tissue lysates were subjected to acetone precipitation; proteins were precipitated at $-20\,^{\circ}$C for 3 h. After centrifugation ($12{,}000 \times g$, 5 min), the pellets were resuspended in 10 ml of 150 mM Tris-HCl, pH 8.0, denatured, and reduced with 20 ml of 9 M urea, 30 mM DTT in 150 mM Tris-HCl, pH 8.0, at 37 °C for 40 min, then alkylated with 40 mM iodacetamide in the dark for 30 min. The reaction mixture was diluted tenfold using 50 mM Tris-HCl pH 8.0 prior to overnight digestion at 37 °C with trypsin (1:20 enzyme to substrate ratio). Digestions were terminated with adding equal volume of 2% formic acid, and then desalted using Waters Oasis HLB 1 ml reverse phase cartridges according to the vendor's procedure. Eluates were dried via vacuum centrifugation.

About 1 µg of the tryptic digest (in 2% acetonitrile/0.1% formic acid in water) was analyzed by LC/MS/MS on an Orbitrap Fusion Tribrid mass spectrometer (Thermo Scientific) interfaced with a Dionex UltiMate 3000 Binary RSLCnano System. Peptides were separated onto an Acclaim PepMap C18 column (75 mm ID × 15 cm, 2 mm) at flow rate of 300 nl/min. Gradient conditions were 3–22% B for 120 min; 22–35% B for 10 min; 35–90% B for 10 min; 90% B held for 10 min, (solvent A, 0.1 % formic acid in water; solvent B, 0.1% formic acid in acetonitrile). The peptides were analyzed using data-dependent acquisition method, Orbitrap Fusion was operated with measurement of FTMS1 at resolutions 120,000 FWHM, scan range 350–1500 $m/z$, AGC target 2E5, and maximum injection time of 50 ms; during a maximum 3 s cycle time, the ITMS2 spectra were collected at rapid scan rate mode, with CID NCE 35, 1.6 $m/z$ isolation window, AGC target 1E4, maximum injection time of 35 ms, and dynamic exclusion was employed for 60 s.

The raw data files were processed using Thermo Scientific Proteome Discoverer software version 1.4, spectra were searched against the Uniprot *Mus musculus* plus DT database, using Sequest HT search engine. Search results were trimmed to a 1% FDR using Percolator. For the trypsin, up to two missed cleavages were allowed. MS tolerance was set 10 p.p.m.; MS/MS tolerance 0.6 Da. Carbamidomethylation on cysteine residues was used as fixed modification; oxidation of methione, as well as phosphorylation of serine, threonine, and tyrosine was set as variable modifications. The quantitative results for brain regions in different distance were calculated based on the absolute LC–MS ion intensity values. For each data point with different distance, the ion intensity score was calculated by the chromatographic peak areas for DT against the total ion area of each sample, and subsequently normalized to tissue weight. So, the relative abundance of DT between samples was obtained[76,77].

**Optogenetics.** For in vivo optogenetic stimulation, the optical fiber (outer diameter 250 µm; core diameter 105 µm; numerical aperture 0.22, FG105LCA, Thorlabs, Newton, NJ) was cut into small pieces (25 mm length), using high precision fiber cleaver (XL411, Thorlabs, Newton, NJ). The optical fiber was glued with ceramic ferrule (inner diameter 230 µm, Kientec System, Stuart, FL) and polished. To target the dlDRN, the optical fiber was installed on the holder and guided into the coordinate ($AP = -4.36$ mm, $ML = \pm 0.35$ mm, $DV = -3.2$ mm). To target the PVN, the optical fiber was inserted with the coordinate ($AP = -0.70$ mm, $ML = +0.2$ mm, $DV = -4.5$ mm). To target the vlPAG the optical fiber was inserted with the coordinate ($AP = -4.60$ mm, $ML = \pm 0.50$ mm, $DV = -2.7$ mm). To target the dmDRN, the optical fiber was implanted with the coordinate ($AP = -4.60$ mm, $ML = \pm 0$ mm, $DV = -3.0$ mm). To perform the photostimulation, the optical fiber was connected to spectralynx (Neuralynx, Inc, USA) through a patch cable. For the assay of temperature measurement in the $Agrp^{Cre}$::Ai32 mice, the blue light was shed into the dlDRN at different frequencies: 0, 5, 10, 15, and 20 Hz with 10 ms pulse for 1 h. For the food intake measurement, the blue light was shed into the PVN or DRN at different frequencies: 0, 10, 15, and 20 Hz with 10 ms pulse for 1 h (6–7 p.m.). The power of laser (0.5–1.2 mW) was calculated by optical power meter (PM100D, Thorlabs, Newton, NJ) before each experiment.

For in vivo optogenetic inactivation, an optical fiber was implanted into the dmDRN with the coordinate ($AP = -4.60$ mm, $ML = \pm 0$ mm, $DV = -3.0$ mm). To perform the photoinhibition, the optical fiber was connected to spectralynx (Neuralynx, Inc, USA) through a patch cable. For the assay of temperature measurement, the $Agrp^{DTR/+}::Mc4r^{Cre}::Sert^{Flp}$ mice were injected with *AAV9-DIO-WGA-Cre-mCitrine* and *AAV9-Con-Fon-eNpHR-mCherry* virus into the dlDRN and dmDRN, respectively. The continuous yellow illumination was shed into the dmDRN from 9 to 10 a.m. The power of yellow light we applied was 0.8 mW at the tip of the fiber, which was calculated by optical power meter (PM100D, Thorlabs, Newton, NJ) before each experiment.

For in vitro optogenetics, an optical fiber (200 µm diameter) was coupled to a 473-nm solid-state laser diode. The fiber was passed through a stainless-steel tube (inner diameter 250 µm, outer diameter 480 µm) and bonded to the tube with glue. The tip of the fiber was trimmed and polished, submerged in artificial cerebrospinal fluid (aCSF), and placed above the dDRN. The blue light was controlled by a pulse stimulator. The blue light pulses (10 ms/pulse, 20 Hz) were shed onto the ChR2-expressing AgRP axonal fibers within the dDRN or ChR2-GFP-labeled dDRN neurons. The power of the laser (0.5–1.2 mW) was measured by a power meter (PM100D, Thorlabs, Newton, NJ) before experiments.

**Drug administration.** To general administration of drugs, the mice were received i. p. injection of CNO at a dose of 1 mg/kg. In some chronic studies, the CNO was i.p. injected at 8 a.m. and 8 p.m. twice per day with a dose of 1 mg/kg up to 7 days.

For drug infusing to brain, the guide cannula (23-gague steel) was obtained from Plastics One (Roanoke, VA). A circular craniotomy (diameter 0.5 mm) was drilled at the locations per different experiments. The guide cannula was installed on the holder and guided into the target brain region. To deliver drug, the internal cannula was inserted onto the top of the guide cannula and extended below the guide cannula 0.5 mm. To target the dlDRN, the cannula was implanted in the dlDRN with the coordinate ($AP = -4.36$ mm, $ML = \pm 0.35$ mm, $DV = -2.8$ mm). AgRP(82–131) (0.6 nM/side, Phoenix pharmaceuticals, Burlingame, CA) or MTII (4 ng, Tocris Bioscience, Minneapolis, MN) was infused between 8 and 9 a.m.

For the study, the effect of SR 59230 A (2 mg/kg/day, Tocris, Minneapolis, MN), a selective β-3AR antagonist, on thermogenesis the drug was subcutaneously administered at 9 a.m. for 4 days in thermogenesis assays.

**Energy expenditure**. $O_2$ consumption, $CO_2$ production, heat production, RER, and total activity were monitored by Comprehensive Lab Animal Monitoring System (CLAMS; Columbus Instruments, Columbus, OH)[19]. Mice were acclimatized in the chambers for 48 h prior to data collection. All CLAMS data were collected from Oxymax software version 5.12.

We measured the core and iBAT temperature at 9 a.m. or 9 a.m.–3 p.m. for a 6-h continuous monitoring in all experiments.

To measure the iBAT temperature, we use radio-frequency identification (RFID) technology, passive RFID transponders were implanted subcutaneously. When the passive RFID transponder is within read range, its internal antenna draws energy from the radio waves emitted by the reader. This energy powers the chip, which then sends data back to the reader. For the implantation, the glass-covered, biocompatible temperature transponders (dimension: 2 mm × 14 mm; model: IPTT-300 transponders; BioMedic Data Systems, Seaford, USA) were loaded in a needle applicator device, and sterilized. Then the temperature transponders were implanted subcutaneously in the region between the scapulae in the anesthetic mice. The mice were observed for up to 48 h, and temperature transponders were checked daily for presence and functionality before the start of the experiment.

The core temperature was measured using rectal temperature probes, which is a common method of measuring body temperature in rodents. It involves inserting a small-diameter temperature probe (RET-3, Kent Scientific, Torrington, CT, USA), a resistance temperature detector, through the anus for a depth of >2 cm to yield stable core temperatures. This rectal probe for mice is attached to a TH-5 thermalert monitoring thermometer (Physitemp, Clifton, NJ), so the value of temperature was measured and displayed.

To monitor in vivo real-time temperature, Anipill capsules (ANIPILL system®; Body-cap, Caen, France) were implanted intraperitoneally under isoflurane anesthesia. Before implantation, the Anipill capsules were activated and paired with mice through individual identification numbers. An incision of 15 mm in length was performed in the right iliac fossa under anesthesia. The Anipill capsule was implanted in the intrperitoneal cavity. The body wall and incision were closed by sutures. Continuous data recording started after a recovery period of 2 weeks, with a sampling frequency of 15 min for all studies.

**Food Intake**. For the acute feeding studies in $Agrp^{Cre}::Ai32$ mice, 1-h food intake was measured (from the start of the "lights off" cycle, 6–7 p.m.) 1 h after photostimulation under different parameters with chow diet (5V5R, LabDiet, St. Louis, MO). Food intake (0.5, 1, 2, 4, and 6 h for CNO) was monitored from 8 p.m. to 2 a.m .with chow diet. For the chronic feeding studies, food intake as well as body weight was daily measured between 9 and 10 a.m. up to 12 weeks.

**In vitro electrophysiology**. Animals were subjected to anesthesia, and the handling protocol of the local committee was followed. In most case, we have used the same mouse first for the in vivo experiments, and immediately after for combined electrophysiology and optogenetics in vitro, in order to minimize the number of mice. Mice were deeply anesthetized with isoflurane and transcardially perfused with a modified ice-cold sucrose-based cutting solution (pH 7.3) containing 10 mM NaCl, 25 mM NaHCO₃, 195 mM sucrose, 5 mM glucose, 2.5 mM KCl, 1.25 mM NaH₂PO₄, 2 mM Na-pyruvate, 0.5 mM CaCl₂, and 7 mM MgCl₂, bubbled continuously with 95% $O_2$ and 5% $CO_2$. The mice were then decapitated, and the entire brain was removed and immediately submerged in the cutting solution. Slices (250–300 μm) were cut with a Microm HM 650 V vibratome (Thermo Scientific). Slices containing the DRN were recovered for 1 h at 34 °C and then maintained at RT in artificial cerebrospinal fluid aCSF (pH 7.3) containing 126 mM NaCl, 2.5 mM KCl, 2.4 mM CaCl₂, 1.2 mM NaH₂PO₄, 1.2 mM MgCl₂, 11.1 mM glucose, and 21.4 mM NaHCO₃, saturated with 95% $O_2$ and 5% $CO_2$ before recording. Slices were transferred to a recording chamber and allowed to equilibrate for at least 10 min before recording. The slices were superfused at 34 °C in oxygenated aCSF at a flow rate of 1.8–2 ml/min. GFP-labeled neurons in the dlDRN were visualized using epifluorescence and IR-DIC imaging on an upright microscope (Eclipse FN-1, Nikon) equipped with a moveable stage (MP-285, Sutter Instrument). Patch pipettes with resistances of 3–5 MΩ were filled with intracellular solution (pH 7.3) containing 128 mM K-gluconate, 10 mM KCl, 10 mM HEPES, 0.1 mM EGTA, 2 mM MgCl₂, 0.05 mM Na-GTP, and 0.05 mM Mg-ATP. Recordings were made using a MultiClamp 700B amplifier (Axon Instrument), sampled using Digidata 1440 A and analyzed offline with pClamp software version 10.3 (Axon Instruments). Series resistance was monitored during the recording, and the values were generally <10 MΩ and were not compensated. The liquid junction potential was +12.5 mV and was corrected after the experiment. Data were excluded if the series resistance increased dramatically during the experiment or without overshoot for action potential. Currents were amplified, filtered at 1 kHz, and digitized at 20 kHz. Current clamp was engaged to test neural firing frequency and resting membrane potential at the baseline or in response to blue light, AgRP (0.5 μM), α-MSH (0.5 μM), and NPY (0.2 μM). A neuron was considered depolarized or hyperpolarized if a change in membrane potential was at least 2 mV in amplitude[78]. In some experiments, the aCSF solution also contained 1 μM TTX and a cocktail of fast synaptic inhibitors, AP5 (30 μM; an NMDA receptor antagonist) and CNQX (30 μM; an AMPA receptor antagonist) to block the majority of presynaptic inputs. For the light-evoked EPSC recordings, the internal recording solution contained: 125 mM CsCH₃SO₃; 10 mM CsCl; 5 mM NaCl; 2 mM MgCl₂; 1 mM EGTA; 10 mM HEPES; 5 mM (Mg)ATP; 0.3 mM (Na) GTP (pH 7.3 with NaOH). mEPSC in the DRN neurons was measured in the voltage clamp mode with a holding potential of −60 mV in the presence of 1 μM TTX and 50 μM bicuculline. 4-AP and TTX were used to confirm the evoked EPSC currents are monosynaptic currents. In some experiments, lucifer yellow (100 nM) was added in the pipette solution to identify the recorded neurons. After recording, slices were fixed with 4% formalin in PBS at 4 °C overnight, and then subjected to post hoc identification of the anatomical location of the recorded neuron within the DRN. TpH2 staining was used to confirm the recorded the DRN neurons are 5-HT neurons. To test the effects of NPY on the neural firing of MC4R neurons the NPY (0.5 μM) was puff for 1 s under current clamp. The inhibitory effects of AgRP on the 5-HT neurons were validated by bath perfusion of AgRP under current clamp.

**In vivo fiber photometry**. For in vivo fiber photometry, $Mc4r^{Cre}$ mice were injected AAV9-CAG-FLEX-GCAMP6f into the dlDRN. Animals were allowed to recover for at least 2 weeks before experiments proceeded. To assemble an optic probe, a jacket (~15 mm long) and acrylic buffer were stripped off at the free ends of the two multimode fibers to expose the cladding. Then fiber ends were cleaved off with a fiber cleaver. Use a scissor to cut two pieces of bare fibers. Each of free fibers was inserted into a ceramic ferrule (128 μm hole) and fixed with super glue. The top of ferrule (with fiber end) was polished by polishing sheets (Thorlabs). Under a dissection microscope, the two exposed fiber ends were placed parallel to each other and super glue was applied to the fiber ends 5 mm away from the tips. Two multimode fiber patch cord (outside diameter of 0.9 mm; core/cladding Ø 105/125 mm, Thorlabs) were used for fluorescence excitation and detection, respectively. The 488 nm laser (OBIS LS 100 mW, Coherent, Santa Clara, CA) was used to excite GCaMP6 through the multimode fiber patch cord, and the QE Pro detector (Ocean optics, Largo, FL) was used to collect the photons emitted from the tissue through the multimode detection fiber patch cord. The OceanView software version 1.2 (Ocean optics, Largo, FL) was used to acquire the data. Spectral channel (500–543 nm) was selected for GCaMP6. The in vivo fiber photometry was performed between 9 and 11 a.m. The integrated photon count was used as a measure of intensity. The spectrum data were recorded continuously at 10 Hz sampling frequency. The percentage $\Delta F/F$ was calculated by $100 \times (F - F_{mean})/F_{mean}$, where $F_{mean}$ was the mean fluorescence intensity throughout the entire acquisition fragment. The detection threshold for a fluorescence transient was defined as $\mu + 3\sigma$, where $\mu$ and $\sigma$ were the mean and the standard deviation of the fluorescence baseline period. The fluorescence transients during baseline were randomly sampled (40–45 s) and normalized within animal by the comparison between baseline and manipulation within individual mouse. For the heat map, the recorded data was saved as ASCII files and plotted by MATLAB R2015a (MathWorks, Natick, MA).

**In vivo tetrode recording**. We designed the microdrive model that enabled deliver laser or drug into brain and recording neural activities simultaneously. The in vivo tetrode recording was performed between 9 to 11 a.m. The microdrives were modified on the basis of tetrode microdrives from Neuralynx (Bozeman, MT) which were loaded with one steel cannula (24-Gauge) in the center and seven nichrome tetrodes consisting of four thin wires twined together (STABLOHM 675, California Fine Wire Co., Grover Beach, CA). The cannula positioning 0.3 mm from the tetrode was glued to the middle of the bundle of tetrodes. Tetrode tips were goldplated to reduce impedance to 0.3–0.4 M (tested at 1 kHz). The microdrive was implanted in the dlDRN in the $Agrp^{DTR/+}::Mc4r^{Cre}$ mice with AAV2-EF1a-DIO-hChR2(E123T/T159C)-GFP and AAV2-hSyn-DIO-hM4D(Gi)-mCherry into the dlDRN. After recovery from microdrive implantation, the mouse was connected to a 32-channel preamplifier headstage. All signals recorded from each tetrode were amplified, filtered between 0.3 and 6 kHz, and digitized at 32 kHz through Cheetah software version 5.6.3. The local field potentials were amplified and filtered between 0.1 and 1 kHz. The tetrodes were slowly lowered in quarter-turns of a screw on the microdrive (~60-μm steps). Spikes were sorted using Plexon Offline Sorter software version 4.4.1. Units were separated by the $T$-distribution E–M method, and cross-correlation and autocorrelation analyses were used to confirm unit separation. Clustered waveforms were subsequently analyzed by using NeuroExplorer version 5.033 (Nex Technologies, Colorado Springs, CO). The firing rates were presented with spikes per bin with 10 s interval or spikes per second. The MC4R neurons were identified by the short latencies of evoked spikes accurately following high-frequency photostimulation, as well as the identical waveforms of evoked and spontaneous spikes. The putative 5-HT neurons were identified by the electrophysiological characteristics, including firing rate, firing rhythmicity, and spike distribution, which displayed slow-firing clock-like pattern[52,67,68]. To ablate AgRP → dlDRN circuit the DT (0.4 ng) was infused to the dlDRN using a needle (Hamilton Small Hub RN 33 G, Reno, NV) connected with a 10 μl syringe (Hamilton 700 Microliter, Reno, NV), at a rate of 0.01 μl/min. A total 0.1 μl volume was delivered into the dlDRN. The tetrode recording was performed 3 days after DT injection.

**Cellular respiration rates**. The adipose tissue oxygen consumption rate (OCR) was assessed using freshly isolated mouse iBAT and scWAT. These tissues were

rinsed with XF Assay medium (containing 2 mM pyruvate, 25 mM glucose, 2 mM glutamine, and 1.85 g/l NaCl) and cut into pieces (4 mg). After extensive washing, one piece of tissue was dried, weighed for normalization, and placed into each tissue of a XF24 Islet Capture Microplate (Seahorse Bioscience, Agilent, Santa Clara, CA) in accordance with the manufacturer's instructions. The tissue was immediately covered with the islet capture screen, allowing for free perfusion while minimizing tissue movement, and 500 μl of the assay medium was added. According to previous report, untreated oxygen consumption was measured for 24 min followed by the measurement of OCR after injection of ATP synthase inhibitor oligomycin (20 μg/ml; 40 min), mitochondrial uncoupled FCCP (20 mM; 40 min), and complex I inhibitor rotenone with complex III inhibitor antimycin A (20 mM respectively; 40 min). Data presented for the XF24 experiments are representative of six independent experiments with 12 replicate wells per genotype/treatment within each experiment. Data were collected and analyzed using Seahorse Wave software version 2.4.

**Quantitative real-time PCR**. Frozen WAT and BAT were homogenized in PureZOL RNA isolation reagent (Bio-Rad, Hercules, CA). Total RNA was extracted using an Aurum Total RNA Fatty and Fibrous Tissue Kit following the manufacturer's instructions (Bio-Rad). A total of 1 μg of total RNA was used to synthesize first-strand cDNA using the SuperScript III First-Strand Synthesis System (ThermoFisher Scientific, Waltham, MA). Quantitative PCR reactions were performed using SsoAdvanced Universal SYBR® Green Supermix (Bio-Rad) in a CFX96 Touch Real-Time PCR Detection System (Bio-Rad). The ΔCt method ($2^{-\Delta Ct}$) was used to calculate the relative mRNA expression level of each gene. Specific gene expression was normalized to *Gapdh*. The real-time PCR was performed using TaqMan gene expression assay for *Mc4r*, *Tph2*, *Ucp1*, *Ppargc1a*, *Cidea*, and *Cited1* or using SYBR Green system for *Gapdh*, *Dio2*, *Cox7a1*, *Car4*, *Fgf21*, *Eva1*, and *Scr*. Primer sequences are available in Supplementary Table 1.

**Histology**. Mice were killed and perfused transcardially with ice-cold PBS buffer (pH 7.4) containing 3% (wt/vol) paraformaldehyde (Alfa Aesar) and 1% glutaraldehyde (Sigma, St. Louis, MO). Brains were collected and postfixed overnight under 4 °C in a fixation buffer containing 3% paraformaldehyde. Free-floating sections (20 μm for the ARC sections and 30 μm for other sections) were cut by a microtome (ThermoFisher Scientific, Waltham, MA), and then blocked with 5% (wt/vol) normal donkey serum in 0.1% Triton X-100 (TBST buffer, pH 7.2) for overnight. For each different assay, either goat anti-AgRP (1:500 dilution; sc-18634, Santa Cruz Biotech, Dallas, TX), rabbit anti-Tph2 (1:1000 dilution; ABN60, EMD Millipore, Burlington, MA), chicken anti-GFP (1:400 dilution; A10262, Invitrogen, Waltham, MA), or rabbit anti-Iba1 (1:1000 dilution; PA5-27436, ThermoFisher Scientific, Waltham, MA) was applied to the sections for overnight incubation under 4 °C, followed by 4 × 15-min rinses in the TBST buffer. Finally, sections were incubated with Alex Fluor 488-conjugated donkey anti-rabbit secondary antibody (1:1000 dilution; 711-545-152, Jackson Immunolab, West Grove, PA), or Alex Fluor 488-conjugated goat anti-chicken secondary antibody (1:1000 dilution; A-11039, ThermoFisher Scientific, Waltham, MA), or Alex Fluor Cy3-conjugated donkey anti-rabbit secondary antibody (1:1000 dilution; 711-585-152, Jackson Immunolab, West Grove, PA), or Alex Fluor Cy3-conjugated donkey anti-goat secondary antibody (1:1000 dilution; 705-585-147, Jackson Immunolab, West Grove, PA) or Alex Fluor Cy5-conjugated donkey anti-rabbit secondary antibody (1:1000 dilution; 711-175-152, Jackson Immunolab, West Grove, PA) for 2 h at RT, followed by 4 × 15-min rinses in TBST buffer. For mounted sections, fluorescent images were captured by a digital camera mounted on an Axio Observer microscope (Zeiss, Thornwood, NY) using AxioVision software version 4.9.1.

H&E staining of paraffin-embedded adipose tissues was performed. Prepare sections of paraffin-embedded adipose tissue samples using a microtome (thickness: 5 μm for WAT, 3 μm for BAT). Transfer the sections into a water bath (~40 °C). Collect the paraffin sections after the paraffin surrounding the tissue has smoothed out. Air-dry the sections before proceeding to the next step. Put the adipose tissue slides in an oven and incubate them for 2 h at 60 °C to remove excess paraffin. Allow the slides to cool to RT. Transfer the slides to a vertical staining jar. Deparaffinize the adipose tissue slides in xylen twice (5 min each). Rehydrate the adipose tissue slides with 99.7, 95, and 70% (vol/vol) ethanol twice (5 min each) under each condition. Wash the adipose tissue slides with dH$_2$O for 5 min. Stain the adipose tissue slides with hematoxylin for 3 min. Remove excess hematoxylin under running water for 10 min. Stain the adipose tissue slides with eosin for 1–2 min. Dehydrate the adipose tissue slides with 95 and 99.7% (vol/vol) ethanol twice (5 min each) under each condition. Remove the adipose tissue slides from the vertical staining jar and allow the slides to dry on the bench. Mount the slides for imaging. The average lipid droplet area and adipocyte size in adipose tissue was measured by using ImageJ software version 1.49 (NIH). To measure the area of the adipocytes, the Measure and Label Macro for ImageJ was installed. This macro provided a read out of the adipocyte area and placed a corresponding unique number within the center of the adipocyte. Using this tool, the individual adipocytes of interest were selected. A unique number was placed in the center of the adipocyte, and the corresponding area was appeared. All of the adipocytes were counted and calculated.

**Statistics and reproducibility**. All statistical results are presented as mean ± s.e.m. Statistical analyses were performed using Graphpad Prism 7.00. Two-tailed Student's *t* tests were used to calculate *P* values of pair-wise comparisons. Data for comparisons across more than two groups were analyzed using a one-way ANOVA followed by Tukey post hoc test. Time course comparisons between groups were analyzed using a two-way ANOVA followed by Bonferroni post hoc test for multiple comparisons. Data were considered significantly different when probability value was <0.05. Immunofluorescence staining, viral expression, or viral tracing were conducted two to three times to ensure reproducibility.

**Reporting summary**. Further information on research design is available in the Nature Research Reporting Summary linked to this article.

## Data availability
The *Mus musculus* database from Uniprot (https://ftp.uniprot.org/pub/databases/uniprot/current_release/knowledgebase/reference_proteomes/Eukaryota/UP000000589/UP000000589_10090.fasta.gz) was used, then inserted the Diphtheria toxin protein fasta file (https://www.uniprot.org/uniprot/Q5PY51.fasta) into it and formed the Uniprot *Mus musculus* plus DT database. The data that support the findings of this study are available from the corresponding author upon reasonable request. Source data are provided with this paper.

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

## Acknowledgements

We appreciate the helpful, comprehensive, comments on the manuscript by Richard Palmiter and associates at the University of Washington (UW). The *Mc4r^Cre* mice were gifts kindly

provided by Bradford Lowell (Beth Israel Deaconess Medical Center, Harvard Medical School). The $Mc4r^{lox}$ and $Mc4r^{loxTB}$ mice were gifts kindly provided by Joel Elmquist (University of Texas Southwestern Medical Center). The $AAV\text{-}DIO\text{-}ChR2$ backbone plasmid was a gift kindly provided by Scott Sternson (HHMI Janelia Research Campus). The $Pet1^{Cre}$ and $Pet1^{CreER}$ mice were gifts kindly provided by Evan Deneris (Case Western Reserve University). The $Sert^{Flp}$ mice were gifts kindly provided by YuFu (National University of Singapore). The lentiviral vector $Ucp1^{shRNA}$ were gifts kindly provided by Shingo Kajimura (University of California San Francisco). The Ai14 and Ai32 mice were gifts kindly provided by Hongkui Zeng (Allen Institute for Brain Sciences). We thank the Alkek Center for Molecular Discovery, the Metabolomics Core, the NMR and Drug Metabolism Core, and the Mouse Metabolism Core at Baylor College of Medicine for providing technical support. The AAV vectors applied in this study were packaged by the Optogenetics and Viral Design/ Expression Core at Baylor College of Medicine. This work was supported by NIH grants (1R01DK109194 and 1R56DK109194) to Q.Wu, the Pew Charitable Trust awards (0026188) to Q.W., American Diabetes Association awards (#7-13-JF-61) to Q.W., Baylor Collaborative Faculty Research Investment Program grants to Q.W., USDA/CRIS grants (3092-5-001-059) to Q.W., the Faculty Start-up grants from USDA/ARS and University of Iowa to Q.W., NIH grants (R01DK093587, R01DK101379, and R01DK117281) to Y.X., USDA/CRIS grants (3092-5-001-059) to Y.X., American Heart Association awards (17GRNT32960003) to Y.X., American Diabetes Association (1-17-PDF-138) to Y.He, Charles H. Hood Foundation to M. O., Dietrich, Whitehall Foundation to M.O., Dietrich, NIH grants (R01DK107916 and P30 DK045735) to M.O., Dietrich, NIH grants (R01DK116899) to M.H.-C., and USDA/CRIS grants (3092-5-001-059) to M.H.-C. This work is supported in part by the Clinical and Translational Proteomics Service Center at the University of Texas Health Science Center. Q. W. is the Pew Scholar of Biomedical Sciences and the Kavli Scholar.

## Author contributions

Conceptualization, Q.W., Y.Ha., G.X., D.S., and F.M.; methodology, Y.Ha., G.X., D.S., F.M., Y.L.H., Y.He., Y.R., M.F., G.H., I.F., and M.H.C.; validation, Q.W., and Y.Ha.; formal analysis, Y.Ha., G.X., and D.S.; investigation, Y.H., G.X., D.S., F.M., Y.L.H., Y.He., Y.R., and M.H.C.; writing—original draft, Q.W., Y.Ha., G.X., and D.S.; funding acquisition, Q.W., M.O.D., and Y.X., and M.H.C.; resources, Q.W., I.F., M.O.D., M.H.C., and Y.X.; and supervision, Q.W.

## Competing interests

The authors declare no competing interests.
