## [Peer Review File · Nature Communications]

Reviewer comments, first round –

Reviewer #1 (Remarks to the Author):

This study reports a circuit from hypothalamic AGRP neurons to MC4r-expressing glutamatergic cells in the dIDRN which themselves activate a subset of mDRN serotonergic neurons. This is an extremely comprehensive study using a variety of cutting edge genetic and neuroscience techniques. I think the major claims of the manuscript are supported by the experiments and the remaining questions are primarily clarifications that need to be addressed. However, despite my admiration for most of the experiments, the text of the manuscript is extremely hard to follow in places. It probably took 3-times longer than it should have for me to understand the experiments that were reported here. This is in addition to a large number of minor errors that I will leave to the copy editors. The impact of this work would be increased if the purpose of experiments and techniques used were better explained in the text.

Major concerns:

1. Fig. 2e-h. Is it only the ZsGreen cells being patched here- this is unclear in the text? Is there any evidence of fast synaptic transmission between AGRP neurons and dIDRN neurons? The characterization of the AGRP neuropeptide on the dIDRN neurons is surprisingly superficial given all of the other details (only one example cell showing a weak transient effect with no control). This needs to be expanded. NPY could also be contributing to the inhibition and this should be checked.
2. Figure 4 is poorly explained and potentially poorly performed, but it was unclear. How were Chr2-expressing neurons defined based on electrical response to blue light (for example see UChida and Cohen Nature 2012 as well as Lima and Zador PLOS1 2009 for a good approach. In Fig 4f there appears to be only a general increase in firing but no phase locked responding to the 20Hz photostimulus. The recordings were in Mc4R-cre mice, so how were 5-HT neurons identified? I am skeptical about all of the claims regarding cell type specificity in this section. If these experiments were not performed with rigorous criteria for establishing cell type identity, then the claims in this section will need to be restated to be more representative of the nature of the experiment.
3. Line 238-252 and associated context is just confusing. For example: "In all postsynaptic dmDRN neurons of MC4RdIDRN neurons the 5-HTdmDRN←dIDRN neurons accounted for 27.6%." I don't know what this sentence means and, also, why the backwards arrow here? Does it mean dIDRN->dmDRN(5HT)?
4. There are many more instances where clarity could be improved and the authors should make an effort to do a better job here.

Minor concerns:

5. Title: Why deciphering? This word implies something about language or codes.
6. In the Introduction on lines 62 there are statements indicating uncertainty ("indicates" on line 62) whether there is a relationship of homeostatic systems between hypothalamus and hindbrain. However, the citations show that this relationship has been established. I think on line 65 this could also be corrected by replacing "this link" with "an additional link"
7. Line 100: Should be Supp Fig 4a
8. Lines 182 and 188: the levels are not normalized. They are larger than the controls even if these differences are not statistically different, although it is not clear whether this statistical test was even performed, so I am not even sure if the statements are statistically true.
9. In addition, page and figure numbers would be appreciated in the future.

Reviewer #2 (Remarks to the Author):

This manuscript by Han et al. proposes a novel, multi-synaptic mechanism through which hypothalamic AgRP neurons regulate energy expenditure. The claim is that this is mediated by a direct arcuate AgRP neuron projection to melanocortin 4-receptor-positive neurons in the dorsal raphe nucleus (DRN), which in turn activate DRN serotonin neurons, upon loss of local AgRP signaling. The findings are quite novel and the work presents some exciting data on mechanisms through which the DRN may regulate energy expenditure through thermogenesis. Furthermore, these findings are of significant interest to the fields of neuroscience and metabolism. The data on local genetic manipulations in Figure 5 of MC4R in the DRN are particularly exciting and believable, and the manuscript would be fitting for Nature Communications. However, a number of key issues, such as claims of specificity, proper controls, presentation of the data, and clarity of the writing would need to be addressed prior to acceptance of the manuscript. Due to the SARS-CoV2 pandemic, it is understandable that it might be exceptionally difficult, if not impossible, to perform additional experiments at this time. There are thus no new added/suggested experiments. It is recommended that the authors do address the major and minor comments (below), to put this manuscript into a form suitable for publication.

Major comments:

1. Claims of specificity – the authors suggest that AgRP projections to the dDRN (and not the vIPAG, for example) are responsible for mediating the proposed effects. However, this is not clearly shown. The authors do demonstrate some degree of specificity to the DRN, but it is highly improbable that these manipulations selectively targeted the dDRN but not the vIPAG (frankly, a semantic difference, depending on which mouse atlas is used). The authors also do not offer coordinates for the vIPAG in the manuscript, and A/P axes are not shown for manipulations to distinguish between differential effects on these two loci (some slices are more rostral, and some more caudal). Altering these claims does not at all diminish the novelty of this manuscript, and would greatly improve it (without requiring new experiments). The same goes for the claim concerning dmDRN serotonin neurons. Why are the authors claiming that these neurons are selectively mediating the downstream effect, when it is clear that there are ventral DRN neurons that are also downstream of DRN MC4R neurons?
2. Proper controls – the authors frequently add extraneous data, which are not properly controlled (and are, in many cases, not relevant to the claims of the paper anyhow). For example, in Figs. 3 and 4, the authors perform calcium and tetrode recordings, respectively, which suggest effects on the baseline activity of MC4R neurons (and, questionably, 5-HT neurons). They then co-administer CNO to demonstrate inhibition of MC4R neurons in various states (e.g. AgRP ablation, cold, thermoneutrality, etc.). However, these results add no new information or validation to the manuscript (and they are, in fact, not controlled properly in a 2x2 design – i.e. with saline/vehicle treatment). It is recommended that the authors compare across conditions, while omitting the CNO treatment component. This will open up more space for clarifying schematics or more detailed figure legends. Many proper controls are also missing from Fig. 7. This figure is also least important to the paper's central thesis.
3. Presentation of the data – there are a number of examples where it is unclear if replicates are biologic or technical, paired or unpaired, etc. Some specific examples of this are in Fig. 6. For example, the Veh/CNO studies in Fig. 6b,d-i,o.
4. Clarity of writing – the manuscript is often quite difficult to read throughout. There are some misattributions of figures (e.g. line 100 should be "Supplementary Fig. 4a" not "Fig. 4a"), the text and legends do not always explain a full figure panel, and there are numerous spelling/grammatical errors (e.g. "thermos-sensing" in the abstract). There are also limited technical and experimental details (when time points were taken, and so on), which need to be added in the main or supplemental text.
5. Potential mechanisms through which AgRP ablation regulates DRN function – the authors show effects of AgRP, but critically do not show any data on alpha-MSH bath application, on MC4R neurons (at least, I could not find any data on this in the manuscript, except possibly by inference in Fig. 2h, comparing baselines of control versus 300nM alpha-MSH). However, they also show a

clear effect of alpha-MSH on DRN 5-HT neurons. This discrepancy should be addressed/clarified. Do the authors have data directly showing an effect of alpha-MSH on DRN MC4R neurons (or data from those same neurons that are inhibited by AgRP neuron terminal stimulation?)? If not, why do they believe they see an effect on DRN 5-HT neurons? These potential mechanisms also deserve more attention in the discussion, where the MC4R KO/restoration studies (in the setting of AgRP presence/ablation) could be addressed. I assume that the authors are suggesting that loss of AgRP neurons here leads to unopposed action of alpha-MSH in the DRN, upon loss of AgRP (which would nicely explain the MC4R KO phenotype)?

Minor comments:

Fig. 2: Panels s-w are very hard to interpret (as noted above). No data directly display excitatory effects of alpha-MSH on DRN MC4R neurons. However, there appears to be a direct effect of alpha-MSH on DRN 5-HT neurons. How do the authors reconcile this? Also, does AgRP inhibit DRN 5-HT neurons (as it directly inhibits DRN MC4R neurons)? It also appears that there is an error or extra panel in Fig. 2v that should be either explained in the figure legend or corrected in the figure (i.e. the first two pairwise data sets are labeled the same, but have different results).

Fig. 3: In panels g-h,k-l, the claims should be that calcium transients are more "dynamic," "variable," or "fluctuating," not necessarily at increased baseline activity (because there's no comparison of different treatments, within groups across time). In Fig. 3m, why do the authors believe they see a decrease in iBAT temp? Under cold challenge, shouldn't iBAT temperature increase? (This does not happen in vehicle animals, as it should.)

Fig. 4: DRN 5-HT neurons are "electrophysiologically" identified? This needs to be elaborated on, as previous studies relying on electrophysiological signature to identify DRN 5-HT neurons have proven unreliable. At the very least, the authors should discuss this caveat in the manuscript.

Reviewer #1 (Remarks to the Author):

This study reports a circuit from hypothalamic AGRP neurons to MC4r-expressing glutamatergic cells in the dIDRN which themselves activate a subset of mDRN serotonergic neurons. This is an extremely comprehensive study using a variety of cutting edge genetic and neuroscience techniques. I think the major claims of the manuscript are supported by the experiments and the remaining questions are primarily clarifications that need to be addressed. However, despite my admiration for most of the experiments, the text of the manuscript is extremely hard to follow in places. It probably took 3-times longer than it should have for me to understand the experiments that were reported here. This is in addition to a large number of minor errors that I will leave to the copy editors. The impact of this work would be increased if the purpose of experiments and techniques used were better explained in the text.

Major concerns:

1. Fig. 2e-h. Is it only the ZsGreen cells being patched here- this is unclear in the text? Is there any evidence of fast synaptic transmission between AGRP neurons and dIDRN neurons? The characterization of the AGRP neuropeptide on the dIDRN neurons is surprisingly superficial given all of the other details (only one example cell showing a weak transient effect with no control). This needs to be expanded. NPY could also be contributing to the inhibition and this should be checked.

Thanks for your comments. We recorded the ZsGreen neurons in Fig. 2e-h. We clarified this in our updated manuscript. Per your suggestion, we further examined the effects of AgRP on the dIDRN neural circuit. Our new *in vitro* recording data indicated that AgRP robustly suppressed the neural activities of 5-HT neurons within the dmDRN (Supplementary Fig. 14). To reveal the potential effect of NPY on MC4R neurons we performed *in vitro* patch-clamp recording. Our data showed that NPY elicit weaker inhibition upon the activities of MC4R neurons as comparing to the effects of AgRP (Supplementary Fig. 12), suggesting that AgRP is the major functional signaling pathway within the AgRP-DRN thermogenesis circuit.

2. Figure 4 is poorly explained and potentially poorly performed, but it was unclear. How were Chr2-expressing neurons defined based on electrical response to blue light (for example see UChida and Cohen Nature 2012 as well as Lima and Zador PLOS1 2009 for a good approach. In Fig 4f there appears to be only a general increase in firing but no phase locked responding to the 20Hz photostimulus. The recordings were in Mc4R-cre mice, so how were 5-HT neurons identified? I am skeptical about all of the claims regarding cell type specificity in this section. If these experiments were not performed with rigorous criteria for establishing cell type identity, then the claims in this section will need to be restated to be more representative of the nature of the experiment.

We appreciate these valuable comments. We added more details about how the Chr2 neurons were picked and validated following the previous studies¹. First, we optogenetically evoked the action potentials under high-frequency photostimulation (Supplementary Fig. 18a, b). The evoked spikes precisely followed the laser pulse within 1-ms latency. Second, we compared the evoked waveform with the spontaneous spikes (Supplementary Fig. 18c). Only the units showing evoked waveforms matching the spontaneous spikes were identified as the Chr2 neurons.

3. Line 238-252 and associated context is just confusing. For example: “In all postsynaptic dmDRN neurons of MC4R^{dIDRN} neurons the 5-HT^{dmDRN}←dIDRN neurons accounted for 27.6%.” I don’t know what this sentence means and, also, why the backwards arrow here? Does it mean dIDRN->dmDRN(5HT)?

Thanks for your comments. To rule out any confusion, we updated text by adopting “5-HT^{dmDRN} neurons receiving projections from MC4R^{dIDRN} neurons”.

4. There are many more instances where clarity could be improved and the authors should make an effort to do a better job here.

Thanks for your comments. We thoroughly revised the manuscript as to improve the clarity and readability of our manuscript.

Minor concerns:

5. Title: Why deciphering? This word implies something about language or codes. In several previously published papers, we adopt this word to suggest the identification of a novel neural circuit.

6. In the Introduction on lines 62 there are statements indicating uncertainty (“indicates” on line 62) whether there is a relationship of homeostatic systems between hypothalamus and hindbrain. However, the citations show that this relationship has been established. I think on line 65 this could also be corrected by replacing “this link” with “an additional link”

Thanks for the comments. We updated the sentences in the revised manuscript.

7. Line 100: Should be Supp Fig 4a

We corrected the error in our revised manuscript.

8. Lines 182 and 188: the levels are not normalized. They are larger than the controls even if these differences are not statistically different, although it is not clear whether this statistical test was even performed, so I am not even sure if the statements are statistically true.

Thanks for your comments. We updated the details about the method of photometry. We performed the calcium imaging and analyzed data according the protocols from the paper of Cui, G. et al. (Nat Protoc. 2014, (9) 1213-1228). The data showed in the original manuscript are normalized within animal. It is the comparison between baseline and manipulation within individual mouse. We added more details in the Methods to clarify the issues.

9. In addition, page and figure numbers would be appreciated in the future.

We added the page and figure numbers in the revised manuscript.

Reviewer #2 (Remarks to the Author):

This manuscript by Han et al. proposes a novel, multi-synaptic mechanism through which hypothalamic AgRP neurons regulate energy expenditure. The claim is that this is mediated by a direct arcuate AgRP neuron projection to melanocortin 4-receptor-positive neurons in the dorsal raphe nucleus (DRN), which in turn activate DRN serotonin neurons, upon loss of local AgRP signaling. The findings are quite novel and the work presents some exciting data on mechanisms through which the DRN may regulate energy expenditure through thermogenesis. Furthermore, these findings are of significant interest to the fields of neuroscience and metabolism. The data on local genetic manipulations in Figure 5 of MC4R in the DRN are particularly exciting and believable, and the manuscript would be fitting for Nature Communications. However, a number of key issues, such as claims of specificity, proper controls, presentation of the data, and clarity of the writing would need to be addressed prior to acceptance of the manuscript. Due to the SARS-CoV2 pandemic, it is understandable that it might be exceptionally difficult, if not impossible, to perform additional experiments at this time. There are thus no new added/suggested experiments. It is recommended that the authors do address the major and minor comments (below), to put this manuscript into a form suitable for publication.

Thanks for your thoughtful consideration. We appreciate these positive comments.

Major comments:

1. Claims of specificity – the authors suggest that AgRP projections to the dIDRN (and not the vIPAG, for example) are responsible for mediating the proposed effects. However, this is not clearly shown. The authors do demonstrate some degree of specificity to the DRN, but it is highly improbable that these manipulations selectively targeted the dIDRN but not the vIPAG (frankly, a semantic difference, depending on which mouse atlas is used). The authors also do not offer coordinates for the vIPAG in the manuscript, and A/P axes are not shown for manipulations to distinguish between differential effects on these two loci (some slices are more rostral, and some more caudal). Altering these claims does not at all diminish the novelty of this manuscript, and would greatly improve it (without requiring new experiments). The same goes for the claim concerning dmDRN serotonin neurons. Why are the authors claiming that these neurons are selectively mediating the downstream effect, when it is clear that there are ventral DRN neurons that are also downstream of DRN MC4R neurons?

Thanks for your comments. We located each of the brain nuclei based upon the Atlas of the Mouse Brain (Franklin and Paxinos). The coordinate of vIPAG and other mentioned brain regions were added in the revised Methods. To reveal the anatomical and functional distinction of the AgRP^{ARC}→dIDRN projections from the nearby vIPAG projections, we performed both anterograde and retrograde tracing studies as summarized below:

- 1) We ablated a subpopulation of AgRP neurons by microinjection of DT into the dIDRN of *AgRP^{Cre/DTR::Ai14}* mice. Ablation of AgRP^{ARC}→dIDRN neurons abolished the axonal fibers in the dIDRN, but the integrity of AgRP fibers in the vIPAG were unchanged (Supplementary Fig. 8)

- 2) We injected the Cre-dependent trans-synaptic *AAV-DIO-WGA-ZsGreen* into the ARC of *Agrp^{Cre}* mice. The tracing data showed that majority of the ZsGreen-positive neurons are located within the dIDRN (Supplementary Fig. 10)
- 3) We labeled the $AgRP^{ARC \rightarrow dIDRN}$ neurons by injecting *HSV-hEF1 α -LSL-hM3Dq-mCherry* into the dIDRN of *Agrp^{Cre}::Npy^{GFP}* mice. The retrograde tracing data showed that the projections from AgRP neurons to the dIDRN showed none collateral projections to other downstream regions (Supplementary Fig. 3a-j)
- 4) We performed optogenetic experiment to examine whether the $AgRP \rightarrow vIPAG$ circuits mediate metabolic phenotypes. Photostimulation within the vIPAG axonal terminals in the *Agrp^{Cre}::Ai32* mice showed no significant change in core body temperature or iBAT temperature (Supplementary Fig. 2d-e).

Together, our results suggest that the $AgRP^{ARC \rightarrow dIDRN}$ projections are anatomically and functionally distinct from the $AgRP^{ARC \rightarrow vIPAG}$ projections.

2. Proper controls – the authors frequently add extraneous data, which are not properly controlled (and are, in many cases, not relevant to the claims of the paper anyhow). For example, in Figs. 3 and 4, the authors perform calcium and tetrode recordings, respectively, which suggest effects on the baseline activity of MC4R neurons (and, questionably, 5-HT neurons). They then co-administer CNO to demonstrate inhibition of MC4R neurons in various states (e.g. AgRP ablation, cold, thermoneutrality, etc.). However, these results add no new information or validation to the manuscript (and they are, in fact, not controlled properly in a 2x2 design – i.e. with saline/vehicle treatment). It is recommended that the authors compare across conditions, while omitting the CNO treatment component. This will open up more space for clarifying schematics or more detailed figure legends. Many proper controls are also missing from Fig. 7. This figure is also least important to the paper's central thesis.

We used *in vivo* photometry and *in vivo* tetrode recording in order to reveal how the $MC4R^{dIDRN}$ neurons respond to ambient temperature and activities of AgRP neurons. To maintain the significance while eliminating extra redundancy, we updated manuscript by keeping the *in vivo* tetrode recording data with additional control groups (Supplementary Fig. 19). We also updated the Fig.6 with new origination.

3. Presentation of the data – there are a number of examples where it is unclear if replicates are biologic or technical, paired or unpaired, etc. Some specific examples of this are in Fig. 6. For example, the Veh/CNO studies in Fig. 6b,d-i,o.

We added more control data and reorganized the data presentation in the new Fig 5a-g, m-n to improve the clarity and significance. These new data clearly showed that the DRN Pet1 neurons play a role in control of thermogenesis.

4. Clarity of writing – the manuscript is often quite difficult to read throughout. There are some misattributions of figures (e.g. line 100 should be “Supplementary Fig. 4a” not “Fig. 4a”), the text and legends do not always explain a full figure panel, and there are numerous spelling/grammatical errors (e.g. “thermos-sensing” in the abstract). There are also limited technical and experimental details (when time points were taken, and so on), which need to be

added in the main or supplemental text.

Thanks for your comments. We corrected all these errors in the revised manuscript.

5. Potential mechanisms through which AgRP ablation regulates DRN function – the authors show effects of AgRP, but critically do not show any data on alpha-MSH bath application, on MC4R neurons (at least, I could not find any data on this in the manuscript, except possibly by inference in Fig. 2h, comparing baselines of control versus 300nM alpha-MSH). However, they also show a clear effect of alpha-MSH on DRN 5-HT neurons. This discrepancy should be addressed/clarified. Do the authors have data directly showing an effect of alpha-MSH on DRN MC4R neurons (or data from those same neurons that are inhibited by AgRP neuron terminal stimulation)? If not, why do they believe they see an effect on DRN 5-HT neurons? These potential mechanisms also deserve more attention in the discussion, where the MC4R KO/restoration studies (in the setting of AgRP presence/ablation) could be addressed. I assume that the authors are suggesting that loss of AgRP neurons here leads to unopposed action of alpha-MSH in the DRN, upon loss of AgRP (which would nicely explain the MC4R KO phenotype)?

We appreciated these valuable comments. Our data showed that treatment of α -MSH significantly potentiated the spontaneous firing activity of ~87% of those post-synaptic neurons (baseline level: 1.8 Hz vs 2.4 Hz) indicating MC4R^{dlDRN} neurons could be regulated by α -MSH (Fig. 2f-i). We further perform patch-clamp recording to demonstrate the role of α -MSH on MC4R^{dlDRN} neurons. The results showed that α -MSH could significantly enhance the firing of MC4R^{dlDRN} neurons from 1.9 Hz to 4.4 Hz (Supplementary Fig. 13).

To establish whether α -MSH enhanced the firing of 5-HT^{dmDRN} neurons directly or indirectly, a cocktail of TTX, CNQX, AP5 and bicuculline was applied to block presynaptic inputs during the recording. The results showed that the 5-HT^{dmDRN} neurons could not respond to α -MSH after blocking the glutamatergic inputs of 5-HT^{dmDRN} neurons (Fig. 2s, t, v, w). These results demonstrate that the α -MSH enhanced the firing of 5-HT^{dmDRN} neurons through indirectly action on the upstream MC4R-expressing neurons. Meanwhile the 5-HT^{dmDRN} neurons received monosynaptic glutamatergic inputs from MC4R^{dlDRN} neurons (Fig. 2n-r). These results suggest that α -MSH enhance neural activities of MC4R^{dlDRN} neurons, which in turn excite 5-HT^{dmDRN} neurons.

Minor comments:

Fig. 2: Panels s-w are very hard to interpret (as noted above). No data directly display excitatory effects of alpha-MSH on DRN MC4R neurons. However, there appears to be a direct effect of alpha-MSH on DRN 5-HT neurons. How do the authors reconcile this? Also, does AgRP inhibit DRN 5-HT neurons (as it directly inhibits DRN MC4R neurons)? It also appears that there is an error or extra panel in Fig. 2v that should be either explained in the figure legend or corrected in the figure (i.e. the first two pairwise data sets are labeled the same, but have different results).

As discussed in our previous responses the data collectively showed that the α -MSH regulate the firing of 5-HT^{dmDRN} neurons indirectly through the MC4R^{dlDRN} neurons. To further show the effects of AgRP on the DRN we added additional *in vitro* data showing that AgRP also robustly suppressed the neural activities of 5-HT neurons in the DRN (Supplementary Fig. 14). Blocking the presynaptic inputs of 5-HT^{dmDRN} neurons AgRP could not inhibit the activities of 5-HT^{dmDRN}

neurons, indicating that AgRP inhibit neural activities of 5-HT^{dmDRN} neurons through suppression of MC4R^{dIDRN} neurons.

In Fig. 2v we totally recorded 28 5-HT^{dmDRN} neurons. Firstly, we applied TTX and CNQX to block the AMPA/kainate receptors. There were 18 5-HT^{dmDRN} neurons (the first pairwise data) could not be activated by α -MSH which means α -MSH could excite the 18 neurons by AMPA/kainate receptors. To test the left 10 5-HT^{dmDRN} neurons, the AP5 was added during ephys recording. The results showed after blocking the AMPA/kainate and NMDA receptors, α -MSH failed to excite the 10 neurons (the second pairwise data). Therefore, α -MSH excited the 10 neurons by NMDA receptors. We have explained this part in the figure legend.

Fig. 3: In panels g-h,k-l, the claims should be that calcium transients are more “dynamic,” “variable,” or “fluctuating,” not necessarily at increased baseline activity (because there’s no comparison of different treatments, within groups across time). In Fig. 3m, why do the authors believe they see a decrease in iBAT temp? Under cold challenge, shouldn’t iBAT temperature increase? (This does not happen in vehicle animals, as it should.) Thanks for your comments. We have updated the description with more accurate words per reviewer’s suggestion.

We measured the iBAT temperature under cold challenge following the protocol as described previously^{2,3}. Briefly, the iBAT temperature was monitored using biocompatible temperature transponders (dimension: 2 mm \times 14 mm; model: IPTT-300 transponders; BioMedic Data Systems) implanted subcutaneously in the region between the scapulae. The mice were maintained under room temperature (23 °C) and then exposed to acute cold (4 °C). Consistent with the findings in literatures, we observed that the iBAT temperature decreased (\sim 1.0 °C) in the mice treated with vehicle within 4 hours of cold exposure.

Fig. 4: DRN 5-HT neurons are “electrophysiologically” identified? This needs to be elaborated on, as previous studies relying on electrophysiological signature to identify DRN 5-HT neurons have proven unreliable. At the very least, the authors should discuss this caveat in the manuscript.

According to the literatures, the 5-HT neurons display diverse *in vivo* spiking behaviors⁴⁻⁷. One subgroup of 5-HT neurons could be identified by electrophysiological characteristics including firing rate, firing rhythmicity, and spike distribution, which displayed slow-firing clock-like pattern⁸⁻¹⁰. In our studies the 5-HT neurons showed the low frequency (1.65 Hz) with a highly regular pattern that was revealed by the narrow interspike interval (Fig. 3j, m). Moreover, 5-HT neurons displayed a pacemaker pattern where autocorrelation histograms typically exhibited two or three regular peaks. All of characteristics indicated that these recorded cells are 5-HT neurons. However, classical electrophysiological identification criteria may misidentify a subpopulation of non-5-HT neurons^{8,10}. In our studies there were 40% putative 5-HT neurons responding to the switch from room temperature to thermoneutral condition (Fig. 3m, o), suggesting that false-positive 5-HT neurons may be inadvertently included by the electrophysiological recording method. A combination of optogenetic and *in vivo* electrophysiological recording approaches may minimize this problem and identify a cohort of 5-HT neurons^{11,12}. We added these discussions to the manuscript accordingly.

References

- 1 Cohen, J. Y., Haesler, S., Vong, L., Lowell, B. B. & Uchida, N. Neuron-type-specific signals for reward and punishment in the ventral tegmental area. *Nature* **482**, 85-88, (2012).
- 2 Bal, N. C. *et al.* Sarcolipin is a newly identified regulator of muscle-based thermogenesis in mammals. *Nat Med* **18**, 1575-1579, (2012).
- 3 Bal, N. C., Maurya, S. K., Singh, S., Wehrens, X. H. & Periasamy, M. Increased Reliance on Muscle-based Thermogenesis upon Acute Minimization of Brown Adipose Tissue Function. *J Biol Chem* **291**, 17247-17257, (2016).
- 4 Kocsis, B., Varga, V., Dahan, L. & Sik, A. Serotonergic neuron diversity: identification of raphe neurons with discharges time-locked to the hippocampal theta rhythm. *Proc Natl Acad Sci U S A* **103**, 1059-1064, (2006).
- 5 Andrade, R. & Haj-Dahmane, S. Serotonin neuron diversity in the dorsal raphe. *ACS chemical neuroscience* **4**, 22-25, (2013).
- 6 Schweimer, J. V., Mallet, N., Sharp, T. & Ungless, M. A. Spike-timing relationship of neurochemically-identified dorsal raphe neurons during cortical slow oscillations. *Neuroscience* **196**, 115-123, (2011).
- 7 Okaty, B. W., Commons, K. G. & Dymecki, S. M. Embracing diversity in the 5-HT neuronal system. *Nat Rev Neurosci* **20**, 397-424, (2019).
- 8 Allers, K. A. & Sharp, T. Neurochemical and anatomical identification of fast- and slow-firing neurones in the rat dorsal raphe nucleus using juxtacellular labelling methods *in vivo*. *Neuroscience* **122**, 193-204, (2003).
- 9 Hajos, M. *et al.* Neurochemical identification of stereotypic burst-firing neurons in the rat dorsal raphe nucleus using juxtacellular labelling methods. *Eur J Neurosci* **25**, 119-126, (2007).
- 10 Schweimer, J. V. & Ungless, M. A. Phasic responses in dorsal raphe serotonin neurons to noxious stimuli. *Neuroscience* **171**, 1209-1215, (2010).
- 11 Liu, Z. *et al.* Dorsal raphe neurons signal reward through 5-HT and glutamate. *Neuron* **81**, 1360-1374, (2014).
- 12 Cohen, J. Y., Amoroso, M. W. & Uchida, N. Serotonergic neurons signal reward and punishment on multiple timescales. *eLife* **4**, (2015).

Reviewer comments, second round –

Reviewer #1 (Remarks to the Author):

The paper from Han, et al is substantially improved and clarified. It is an exceptionally comprehensive study, and it sets a very high standard for the depth of circuit analysis. I hope that this manuscript is well-read by neural circuit researchers because there are many impressive genetic tricks for elucidating circuit connectivity and functions. There is a substantive issue remaining with one of the experiments, but the paper is otherwise suitable for publication with minor changes.

The main issue that needs to be addressed is that the viral strategy reported in Fig5h, is unlikely to work in the way that the authors report. The presence of Cre in the Cre-dependent DIO construct, likely led to loss of the DIO during viral preparation. This would be clear if the AAV prep was injected into a wt mouse (lacking Cre). This figure and the associated experiments could be removed from the paper. Alternatively, the authors need to prove that expression of this AAV prep at the same titer and duration does not show Cre-independent expression, which is unlikely to be the case.

Minor:

1. Line 130 says MC4RdIDRN neurons are completely separated from 5HTdmDRN neurons. Line 135 says these neurons are entangled. Supp Fig 9 shows that they are adjacent and largely, but not completely, spatially separated.
2. Fig S10e should say Tph2, not Thp2
3. Fig S12, Statistica tests should be reported for electrophysiological response to NPY
4. Line 164: 4-aminopyridine is misspelled
5. Fig S16h, S17c—fluorescence is misspelled
6. 5-HT neurons recorded by in vivo electrophysiology need to be described as putative 5-HT neurons in the results (as they are in the methods and discussion) because only firing characteristics were used for the classification.

Reviewer #2 (Remarks to the Author):

This updated manuscript by Han et al. has a number of significant improvements, notably a lot of new data for key control studies, which significantly bolster the claims of the paper. Indeed, the data required to support many of the paper's key claims are contained within the present manuscript. However, there are still some key improvements that can be made to the manuscript, which are outlined below. As noted previously, there are no new experiments requested for this paper to be published. To have this manuscript in publishable form, there needs to be some reorganization of the data's presentation, and with caveats included in the discussion about the conclusions drawn.

Comments:

Presentation and analysis of the data – the main figure of concern, here, is Fig. 3. In this figure, which has been minimally amended since the previous round of review (it was previously Fig. 4), the authors still do not focus their analyses on the key comparison(s) required for the conclusions that they draw. The use of DREADDs in this study is a major confound and does not help to support their claims of the effect of loss of Agrp neuron signaling on the activity of DRN neurons (Mc4r or otherwise). If the authors directly compare firing rates in directly across TN vs. RT, as well as non-ablated vs. ablated (instead of investigating the effects of CNO), then I believe that this data is supportive of the claims being made (i.e. the local function of Agrp neurons, at baseline, is to chronically suppress the activity of DRN neurons, and their loss subsequently leads to a significant elevation in DRN neuron activity). But in its current form, Fig. 3's presentation and analyses do not present the data in a way that clearly supports their conclusions.

Proper controls – Fig. 5h-n is not well-controlled, nor are the experiments in Fig. 6. The experiments in Fig. 6f-q, for example, could just omit the Ucp1 knockdown study, which would not at all detract from the paper in its present form, but would resultantly be properly controlled.

REVIEWER COMMENTS

Reviewer #1 (Remarks to the Author):

The paper from Han, et al is substantially improved and clarified. It is an exceptionally comprehensive study, and it sets a very high standard for the depth of circuit analysis. I hope that this manuscript is well-read by neural circuit researchers because there are many impressive genetic tricks for elucidating circuit connectivity and functions. There is a substantive issue remaining with one of the experiments, but the paper is otherwise suitable for publication with minor changes.

The main issue that needs to be addressed is that the viral strategy reported in Fig5h, is unlikely to work in the way that the authors report. The presence of Cre in the Cre-dependent DIO construct, likely led to loss of the DIO during viral preparation. This would be clear if the AAV prep was injected into a wt mouse (lacking Cre). This figure and the associated experiments could be removed from the paper. Alternatively, the authors need to prove that expression of this AAV prep at the same titer and duration does not show Cre-independent expression, which is unlikely to be the case.

We appreciate these comments. To examine if the AAV9-DIO-WGA-Cre-mCitrine virus works in a Cre-dependent manner, we injected this virus into the dIDRN of *WT* mice using the same titer and performed the histological verification after the same duration as showed in the Fig. 5h. Our result showed no mCitrine expression in the dIDRN, suggesting that the viral vector functions as we expected (Supplementary Fig. 23d).

Minor:

1. Line 130 says MC4R^{dIDRN} neurons are completely separated from 5HT^{dmDRN} neurons. Line 135 says these neurons are entangled. Supp Fig 9 shows that they are adjacent and largely, but not completely, spatially separated.

Thanks for your comments. The MC4R^{dIDRN} neurons and 5-HT^{dmDRN} neurons are adjacent but segregated without overlapping. We have modified in our revised manuscript.

2. Fig S10e should say Tph2, not Thp2

We corrected the error in our revised manuscript.

3. Fig S12, Statisticla tests should be reported for electrophysiological response to NPY

Thanks for your comments. We have added the statistic tests in the legend of Fig S12.

4. Line 164: 4-aminopyridine is misspelled

We corrected the error in our revised manuscript.

5. Fig S16h, S17c—fluorescence is misspelled

We corrected the error in our revised manuscript.

6. 5-HT neurons recorded by in vivo electrophysiology need to be described as putative 5-HT neurons in the results (as they are in the methods and discussion) because only firing characteristics were used for the classification.

Thanks for your comments. We have modified the description as reviewer suggested in the results and figures.

Reviewer #2 (Remarks to the Author):

This updated manuscript by Han et al. has a number of significant improvements, notably a lot of new data for key control studies, which significantly bolster the claims of the paper. Indeed, the data required to support many of the paper's key claims are contained within the present manuscript. However, there are still some key improvements that can be made to the manuscript, which are outlined below. As noted previously, there are no new experiments requested for this paper to be published. To have this manuscript in publishable form, there needs to be some reorganization of the data's presentation, and with caveats included in the discussion about the conclusions drawn.

Comments:

Presentation and analysis of the data – the main figure of concern, here, is Fig. 3. In this figure, which has been minimally amended since the previous round of review (it was previously Fig. 4), the authors still do not focus their analyses on the key comparison(s) required for the conclusions that they draw. The use of DREADDs in this study is a major confound and does not help to support their claims of the effect of loss of Agrp neuron signaling on the activity of DRN neurons (Mc4r or otherwise). If the authors directly compare firing rates in directly across TN vs. RT, as well as non-ablated vs. ablated (instead of investigating the effects of CNO), then I believe that this data is supportive of the claims being made (i.e. the local function of Agrp neurons, at baseline, is to chronically suppress the activity of DRN neurons, and their loss subsequently leads to a significant elevation in DRN neuron activity). But in its current form, Fig. 3's presentation and analyses do not present the data in a way that clearly supports their conclusions.

Thanks for your suggestion. We have updated our manuscript with the new comparisons (RT vs TN, non-ablated vs ablated) in the Fig. 3 which will significantly support our conclusions. Our results showed that both of MC4R neurons and putative 5-HT neurons in the DRN could respond to a shift from a RT to TN environment (Fig. 3i-j, m). And the firing rate of both of MC4R

neurons and putative 5-HT neurons in the DRN were significantly increased after ablation of AgRP neurons that project to DRN (Fig. 3k-m).

Proper controls – Fig. 5h-n is not well-controlled, nor are the experiments in Fig. 6. The experiments in Fig. 6f-q, for example, could just omit the Ucp1 knockdown study, which would not at all detract from the paper in its present form, but would resultantly be properly controlled.

Thanks for your comments. We updated manuscript by adding additional control groups of Fig. 5h-n (Supplementary Fig. 23). We also removed the Ucp1^{shRNA} group in the old Fig. 6f-q as reviewer suggested.

Reviewer comments, third round –

Reviewer #1 (Remarks to the Author):

The authors have addressed my concerns and the paper is now suitable for publication

Reviewer #2 (Remarks to the Author):

The updated manuscript by Han et al. describes exhaustive studies (impressively so) on the role of Agrp input to the DRN. On the balance, there are a lot of fascinating data in this report that represent advances in our understanding of how the DRN regulates energy expenditure.

I would encourage the authors to add a caveat statement about the controls used in Fig5h-n.

Regardless, I believe that this manuscript is now suitable for publication in Nature Communications and does not need another round of revision.

REVIEWERS' COMMENTS

Reviewer #1 (Remarks to the Author):

The authors have addressed my concerns and the paper is now suitable for publication

Thanks for your support on our manuscript.

Reviewer #2 (Remarks to the Author):

The updated manuscript by Han et al. describes exhaustive studies (impressively so) on the role of Agrp input to the DRN. On the balance, there are a lot of fascinating data in this report that represent advances in our understanding of how the DRN regulates energy expenditure.

I would encourage the authors to add a caveat statement about the controls used in Fig5h-n.

Regardless, I believe that this manuscript is now suitable for publication in Nature Communications and does not need another round of revision.

We appreciate your comments. Per your suggestion, we have added a caveat statement in the results of Fig. 5m-n.